# DIFFERENTIABLE DAG SAMPLING

**Bertrand Charpentier, Simon Kibler, Stephan Günnemann**
Department of Informatics & Munich Data Science Institute
Technical University Munich
`{charpent, kibler, guennemann}@in.tum.de`

## ABSTRACT

We propose a new differentiable probabilistic model over DAGs (DP-DAG). DP-DAG allows *fast* and *differentiable* DAG sampling suited to continuous optimization. To this end, DP-DAG samples a DAG by successively **(1)** sampling a linear ordering of the node and **(2)** sampling edges consistent with the sampled linear ordering. We further propose VI-DP-DAG, a new method for DAG learning from observational data which combines DP-DAG with variational inference. Hence, VI-DP-DAG approximates the posterior probability over DAG edges given the observed data. VI-DP-DAG is guaranteed to output a valid DAG at *any* time during training and does *not* require any complex augmented Lagrangian optimization scheme in contrast to existing differentiable DAG learning approaches. In our extensive experiments, we compare VI-DP-DAG to other differentiable DAG learning baselines on synthetic and real datasets. VI-DP-DAG significantly improves DAG structure and causal mechanism learning while training faster than competitors.

## 1 INTRODUCTION

Directed Acyclic Graphs (DAGs) are important mathematical objects in many machine learning tasks. For example, a direct application of DAGs is to represent causal relationships in a system of variables. In this case, variables are represented as nodes and causal relationships are represented as directed edges. Hence, DAG learning has found many applications for causal discovery in biology, economics or planning (Pearl, 1988; Ramsey et al., 2017; Sachs et al., 2005; Zhang et al., 2013). However, DAG learning is a challenging problem for two reasons. First, while DAG learning with data from randomized and controlled experiments is the gold-standard for causal discovery, experimental data might be hard or unethical to obtain in practice. Hence, a more common but also challenging setting is DAG learning from observational data which is possible under proper conditions (Pearl, 2009; Spirtes et al., 2000). Second, the number of possible DAGs scales super-exponentially with the number of variables which makes DAG learning an NP hard problem (Chickering et al., 2012; Robinson, 1973).

A first traditional family of models for DAG learning are discrete score-based approaches. These approaches aim at solving the following discrete optimization problem:

$$\max_{G} \; score(\mathbf{X}, G) \text{ s.t. } G \in \text{ discrete DAGs} \tag{1}$$

where $\mathbf{X}$ denotes the observed data and the discrete DAGs space is composed of DAGs with unweighted (present or absent) edges. Examples of score functions are Bayesian Information Criteria (BIC) (Chickering & Heckerman, 1997) or Minimum Description Length (MDL) (Bouckaert, 1993). Discrete approaches have two main limitations: **(1)** the optimization search space of discrete DAGs is large and constrained which often makes the problem intractable without further assumptions, and **(2)** the learning procedure is not differentiable and thus not amenable to gradient-based optimization, as done by most deep learning approaches. To mitigate these issues, a second more recent family of models for DAG learning proposes to leverage continuous optimization by using an augmented Lagrangian formulation (Lachapelle et al., 2020; Ng et al., 2019; Wehenkel & Louppe, 2021; Yu et al., 2019; Zheng et al., 2018). These approaches aim at solving the following optimization problem:

$$\max_{G} \; score(\mathbf{X}, G) \text{ s.t. } h(G) = 0 \tag{2}$$

where $h(G)$ is a smooth function over weighted graphs $G$ which is equal to zero when the graph $G$ satisfies the DAG constraints. Standard examples of score functions are (negative) cross-entropy (Yu

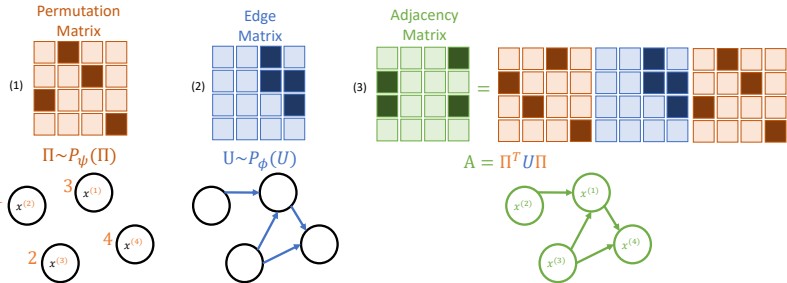

Figure 1: Overview of differentiable DAG sampling.

et al., 2019) or Mean Squared Error (MSE) (Ng et al., 2019) when reconstructing the data. During optimization, $G$ is a weighted graph where the discreteness and acyclicity constraints are relaxed. Hence, these continuous optimization approaches have two main limitations: **(1)** the augmented Lagrangian optimization is computationally expensive as it requires multiple complex dual ascent iterations, and **(2)** the discrete and acyclicity constraints are relaxed during optimization which does not guarantee valid discrete DAGs without non-differentiable pre- and post-processing as proposed by Causal Additive Model (CAM) (Bühlmann et al., 2014). For a more comprehensive description, we recall the augmented Lagrangian optimization method in detail in App. A.

In this paper, we focus on differentiable DAG learning methods and make the following contributions:

- We propose a new probabilistic model over DAGs (DP-DAG) which is capable of *fast* and *differentiable* sampling. DP-DAG can be implemented in few lines of code using Gumbel-Sinkhorn, Gumbel-Top-k and Gumbel-Softmax distributions to parametrize differentiable sampling over permutations and edges (see Fig. 1 and Fig. 2).

- We propose a new method for DAG learning from observational data (VI-DP-DAG) which instantiates a general probabilistic formulation for DAG learning with DP-DAG and variational inference. VI-DP-DAG guarantees *valid* DAG outputs at *any* time during training.

- We show in our experiments on established synthetic and real datasets that DP-DAG outperforms other differentiable DAG learning baselines for DAG structure and causal mechanisms learning while training one order of magnitude faster.

## 2 RELATED WORK

We differentiate between three types of DAG learning approaches: the discrete optimization approaches, the continuous optimization approaches and the sampling-based approaches. We refer to the survey (Vowels et al., 2021) for a more detailed overview of DAG learning approaches.

**Discrete optimization.** First, to make the search space more tractable, discrete optimization approaches modify the original problem with additional assumptions on DAG treewidth (Nie et al., 2014; Scanagatta et al., 2016), ancestral constraints (Chen et al., 2016) or on the number of parents of each variable (Viinikka et al., 2020). Other methods are based on greedy search (Chickering, 2002) or discrete optimization of the topological order (Park & Klabjan, 2017; Scanagatta et al., 2015; Teyssier & Koller, 2005). Another type of discrete optimization approaches are constraint-based methods. These methods explore the discrete DAG space by performing independence tests between observed variables (Bühlmann et al., 2014; Spirtes et al., 2001).

**Continuous optimization.** Second, continuous approaches usually relax the discreteness and acyclicity constraints by using an augmented Lagrangian formulation of the optimization problem (Lachapelle et al., 2020; Ng et al., 2019; Wehenkel & Louppe, 2021; Yu et al., 2019; Zheng et al., 2018). Some approaches define the DAG structure from neural network weights (Lachapelle et al., 2020; Zheng et al., 2018) while other approaches directly learn the DAG adjacency matrix (Ng et al., 2019; Wehenkel & Louppe, 2021; Yu et al., 2019). In contrast to these methods, VI-DP-DAG learns a probabilistic model over the DAG structure. Further, these approaches penalize DAG constraints violation in the augmented Lagrangian formulation but do not guarantee that they are fulfilled during training. Recently, Yu et al. (2021) propose to complement the augmented Lagrangian optimization with a second step projecting the learned graph on admissible solutions. Hence, contrary to VI-DP-DAG, most of these approaches use non-differentiable processing steps – e.g. removing cycles and

spurious edges – to output valid and high-quality DAGs. Common examples of processing steps are Preliminary Neighbors Selection (PNS) and CAM pruning (Bühlmann et al., 2014).

**Sampling-based optimization.** Third, other works use DAG sampling to estimate the posterior distribution over DAGs with MCMC (Kuipers et al., 2020; Niinimäki et al., 2011; 2016; Talvitie et al., 2020; Viinikka et al., 2020). While previous works improve the quality and speed of MCMC computations by sampling (partial) orders or making assumptions on the number of parents per node, they are still computationally extremely expensive (Kuipers et al., 2020; Niinimäki et al., 2011; 2016; Viinikka et al., 2020). E.g., Viinikka et al. (2020) recommend to run MCMC methods during 12 hours to sample from the posterior distribution over DAGs with 100 nodes. In contrast, VI-DP-DAG approximates the posterior distribution over DAG edges with variational inference and can sample very fast. E.g., our VI-DP-DAG trains in around $190$ seconds and samples in less than $1$ second for a DAG with $100$ nodes. Further, while the construction of the MCMC chains are generally non-differentiable, our DP-DAG is capable of fully differentiable DAG learning and can leverage gradient-based optimization. Other works propose optimization of discrete problems using differentiable probabilistic distribution over various discrete objects like subsets or spanning trees but not on DAG structures (Grathwohl et al., 2021; Karalias & Loukas, 2020; Paulus et al., 2020). Further, recent works combine differentiable edge sampling with Gumbel trick and Lagrangian optimization but do not define valid distributions over the full DAG structure (Brouillard et al., 2020; Ng et al., 2019). In contrast, DP-DAG does not require complex Lagrangian optimization and guarantees valid DAGs solutions at any time during training. Finally, Grosse et al. (2021) explores an orthogonal direction where the search space in sequential decision making problems is represented with a DAG.

# 3 PROBABILISTIC MODEL OVER DAGs

A Directed Acyclic Graph (DAG) is a graph $G = (V, E)$ with $n$ nodes $x_1, ..., x_n$ and $m$ directed edges which does not exhibit directed cycles. A DAG always admits a *valid permutation (or linear ordering)* $\pi : [\![1, n]\!] \to [\![1, n]\!]$ of the nodes such that a node cannot have a direct edge toward a node with lower rank i.e., $\pi(i) < \pi(j)$ implies no directed edge from node $x_{\pi(j)}$ to node $x_{\pi(i)}$. Valid permutations are often not unique. Interestingly, this property has an equivalent matrix formulation:

**Theorem 1.** *Lets $\mathbf{A} \in \{0, 1\}^n$ be the (a priori non-triangular) adjacency matrix associated with an arbitrary node labelling of the DAG $G$. The adjacency matrix $\mathbf{A}$ always admits a permutation matrix $\mathbf{\Pi} \in \{0, 1\}^{n \times n}$ and an upper triangular matrix $\mathbf{U} \in \{0, 1\}^{n \times n}$ such that $\mathbf{A} = \mathbf{\Pi}^T \mathbf{U} \mathbf{\Pi}$.*

The *permutation matrix* $\mathbf{\Pi}$ directly corresponds to a valid *component-wise permutation* $\pi$. Hence, Th. 1 simply states that the matrix $\mathbf{U}$ is the new adjacency matrix where the new node labels are a valid permutation of the original node labels i.e. $\mathbf{A}_{ij} = \mathbf{U}_{\pi(i)\pi(j)}$ such that $\mathbf{U}_{\pi(i)\pi(j)} = 0$ if $\pi(i) < \pi(j)$. The decomposition of the DAG adjacency matrix $\mathbf{A}$ in Th. 1 is generally not unique as a DAG $G$ generally admits multiple valid linear permutations $\mathbf{\Pi}$. Hence, we define the following probabilistic model over DAGs (DP-DAG) based on the adjacency matrix decomposition in Th. 1:

$$\mathbb{P}(\mathbf{A}) = \sum_{\mathbf{\Pi} \in \mathcal{P}(G), \mathbf{U} \in \mathcal{U}_n} \mathbb{P}(\mathbf{U}) \, \mathbb{P}(\mathbf{\Pi}) \quad \text{s.t.} \quad \mathbf{A} = \mathbf{\Pi}^T \mathbf{U} \mathbf{\Pi} \tag{3}$$

where $\mathcal{P}(G)$ is the set of valid permutation matrices for the DAG $G$, $\mathcal{U}_n$ is the space of binary upper-triangular matrices of size $n \times n$, $\mathbb{P}(\mathbf{\Pi})$ is the distribution over permutations and $\mathbb{P}(\mathbf{U})$ is the distribution over edges consistent with the sampled permutation of the nodes. Note that the number of valid permutations $|\mathcal{P}(G)|$ can be exponentially large in the number of nodes which makes the exact computation of the probability of a given DAG adjacency matrix $\mathbb{P}(\mathbf{A})$ intractable for large graphs. However, DAG sampling does not require any enumeration of the valid linear permutations. Indeed, we propose a new differentiable DAG sampling method (i.e. $\mathbf{A} \sim \mathbb{P}_{\phi,\psi}(\mathbf{A})$) based on differentiable edge sampling (i.e. $\mathbf{U} \sim \mathbb{P}_\phi(\mathbf{U})$) and differentiable permutation sampling (i.e. $\mathbf{\Pi} \sim \mathbb{P}_\psi(\mathbf{\Pi})$). The variables $\psi$ and $\phi$ denote the parameters of the edge and permutation distributions.

**Differentiable edge sampling.** The Bernoulli distribution is a well-suited distribution to model randomness over a discrete binary variable like an edge $U_{ij} \in \{0, 1\} \sim \text{Ber}(p)$ where $p \in [0, 1]$ is the probability for the edge to exist. Unfortunately, standard random sampling operations from the Bernoulli distribution are not differentiable. In contrast, the Gumbel-Softmax distribution allows for differentiable sampling and approximates the Bernoulli distribution (Jang et al., 2017). The Gumbel-Softmax distribution is defined on continuous variables, i.e. $\hat{U}_{ij} \in [0, 1] \sim \text{Gumbel-Softmax}_\tau(\phi)$

with $\phi \in [0, 1]$, where the temperature parameter $\tau$ allows to interpolate between a one-hot-encoded categorical distribution ($\tau \to 0$) and continuous categorical densities ($\tau \to +\infty$). For differentiable sampling, we can use the straight-through estimator (Bengio et al., 2013): we use the discrete variable $U_{ij} = \arg\max[1 - \hat{U}_{ij}, \hat{U}_{ij}]$ in the forward pass, and the continuous approximation $\hat{U}_{ij}$ in the backward pass. Thus, sampling all the upper triangular indices of $\mathbf{U} \in \{0, 1\}^{n \times n}$ has complexity $\mathcal{O}(n^2)$. We recall the definition of the Gumbel-Softmax distribution in detail in App. B.1.

**Differentiable permutation sampling.** Similarly to an edge, a permutation $\mathbf{\Pi}$ is discrete, making differentiable sampling challenging. We describe two alternative methods which allow for differentiable permutation sampling. First, the Gumbel-Sinkhorn (Mena et al., 2018) is defined on a continuous relaxation of the permutation matrix, i.e. $\hat{\mathbf{\Pi}} \in [0, 1]^{n \times n} \sim \text{Gumbel-Sinkhorn}_\tau(\psi)$ with $\psi \in [0, 1]^{n \times n}$, where the temperature parameter $\tau$ also allows to interpolate between discrete and continuous distributions similarly to the Gumbel-Softmax distribution. For differentiable sampling, we can use the straight-through estimator (Bengio et al., 2013): we use the discrete permutation $\mathbf{\Pi} = \text{Hungarian}(\hat{\mathbf{\Pi}})$ by applying the Hungarian algorithm (Munkres, 1957) to compute a discrete permutation in the forward pass, and the continuous approximation $\hat{\mathbf{\Pi}}$ in the backward pass. Sampling a permutation matrix $\mathbf{\Pi} \in \{0, 1\}^{n \times n}$ is dominated by the Hungarian algorithm and has a complexity of $\mathcal{O}(n^3)$. We recall the definition of the Gumbel-Sinkhorn distribution in detail in App. B.2.

A second method orthogonal to the Gumbel-Sinkhorn method is to use the combination of the Gumbel-Top-k trick (Kool et al., 2019) and the SoftSort operator (Prillo & Eisenschlos, 2020) which also defines a distribution on a continuous relaxation of the permutation matrix. For $k = n$, the Gumbel-Top-n distribution states the sorted perturbed log-probabilities, i.e. $\pi = \text{Sort}(\psi + \mathbf{G})$ where parameters $\psi$ are log-probabilities and $\mathbf{G} \in \mathbb{R}^n$ are i.i.d. Gumbel noise, defines a distribution over *component-wise permutation* (a.k.a. linear ordering without replacement). Instead of the Sort operator, we apply the SoftSort operator to the perturbed log-probabilities which outputs a *continuous relaxation of the permutation matrix*, i.e. $\hat{\mathbf{\Pi}} = \text{SoftSort}(\psi + \mathbf{G}) \in \mathbb{R}^{n \times n}$. For differentiable sampling, we use the straight-through estimator (Bengio et al., 2013): we use the discrete permutation $\mathbf{\Pi} = \arg\max \hat{\mathbf{\Pi}}$ by applying the (one-hot) argmax operator row-wise (Prillo & Eisenschlos, 2020) in the forward pass, and the continuous approximation $\hat{\mathbf{\Pi}}$ in the backward pass. Sampling a permutation matrix $\mathbf{\Pi} \in \{0, 1\}^{n \times n}$ is dominated by the SoftSort operation and has a complexity of $\mathcal{O}(n^2)$. The permutation sampling complexity with Gumbel-Top-k combined with SoftSort is thus lower than the permutation sampling complexity with Gumbel-Sinkhorn. We recall the definition of the Gumbel-Top-k distribution and SoftSort operator in detail in App. B.3 and App. C.

**Differentiable DAG sampling.** Given the aforementioned methods for differentiable edge and permutation sampling, we propose a new simple and valid sampling procedure for DAG sampling which consists in three steps (see Fig. 1): **(1)** Sample a permutation $\mathbf{\Pi}$ from a probabilistic model over permutations $\mathbb{P}_\psi(\mathbf{\Pi})$ i.e. $\mathbf{\Pi} \sim \mathbb{P}_\psi(\mathbf{\Pi})$. **(2)** Sample an upper triangular matrix $\mathbf{U}$ by sampling the upper triangular elements from a probabilistic model over edges $\mathbb{P}_\phi(U_{ij})$ i.e. $U_{ij} \sim \mathbb{P}_\phi(U_{ij})$. **(3)** Compute the final adjacency matrix $\mathbf{A}$ from the permutation matrix $\mathbf{\Pi}$ and the upper triangular matrix $\mathbf{U}$ i.e. $\mathbf{A} = \mathbf{\Pi}^T \mathbf{U} \mathbf{\Pi}$. This procedure is capable of sampling *any* possible DAGs of $n$ nodes because of Th. 1. In practice, we propose to parametrize the distribution $\mathbb{P}_\psi(\mathbf{\Pi})$ using the Gumbel-Sinkhorn or the Gumbel-Top-k trick which define valid distributions over permutations, and parametrize the distributions $\mathbb{P}_\phi(U_{ij})$ using the Gumbel-Softmax trick which defines a valid distribution over edges. Given these parametrizations, the sampling procedure allows *fast* and *differentiable* sampling and can be implemented in a few lines of code (see Fig. 2). The total DAG sampling complexity is dominated by the permutation sampling step which has a complexity of $\mathcal{O}(n^3)$ using Gumbel-Sinkhorn and $\mathcal{O}(n^2)$ using Gumbel-Top-k combined with SoftSort. Finally, the DAG sampling procedure of DP-DAG guarantees a valid DAG output at *any* time during training without additional pre- or post-processing steps.

```python
def differentiable_dag_sample(self):
    # (1) Pi ~ P(Pi) using Gumbel-Sinkhorn or Gumbel-Top-k
    Pi = self.sample_permutation()
    Pi_inv = P.transpose(0, 1)
    # (2) U ~ P(U) using Gumbel-Softmax
    dag_adj = self.sample_edges()
    # (3) A = Pi^T U Pi using Theorem 1
    mask = torch.triu(torch.ones(self.n_nodes, self.n_nodes), 1)
    dag_adj = dag_adj * (Pi_inv @ mask @ Pi)
    return dag_adj
```

Figure 2: Differentiable DAG sampling in Python

## 4 VARIATIONAL DAG LEARNING FROM OBSERVATIONAL DATA

**Structural Equation Model (SEM).** We assume that the data follow a SEM i.e. $x_i = f_i^*(x_{\text{pa}^*(i)}, \epsilon_i)$ where $\epsilon_i$ are independent noises and $f_i^*$ is the ground-truth causal mechanism which deterministically transforms the parent causes $x_{\text{pa}^*(i)}$ and noise $\epsilon_i$ in the consequence value $x_i$. We assume that all variables are observed. We aim at learning both the true parents $\text{pa}^*(i)$ for each node $i$ describing the direct cause-consequence relationship, and the true causal mechanisms $f_i^*$. Alternatively, a SEM can equivalently be written in a matrix form. Indeed, one can use the DAG adjacency matrix $\mathbf{A}^*$ as a mask before applying the causal mechanism $f_i^*$ i.e. $X_i = f_i^*(\mathbf{A}_i^* \circ \mathbf{X}, \epsilon_i)$. Similar masking formulations of a SEM have been used in previous works (Brouillard et al., 2020; Ng et al., 2019).

**Probabilistic DAG learning loss.** We propose a new general formulation for DAG learning based on differentiable DAG sampling which intuitively aims at maximizing the expected score $\mathbb{E}_G[score(\mathbf{X}, G)]$ under the probability distribution over DAGs $P_{\phi,\psi}(G)$ i.e.:

$$\max_{\phi,\psi} \mathbb{E}_G[score(\mathbf{X}, G)] \text{ s.t. } G \sim \mathbb{P}_{\phi,\psi}(G) \tag{4}$$

This formulation allows a rich probabilistic solution that assigns a confidence score to all possible DAGs. In contrast, the discrete and continuous DAG learning formulations in Eq. 1 and Eq. 2 only compute a single point estimate DAG solution and do not model any uncertainty on the final DAG output. A specific instance of the Eq. 4 is the optimization of the following ELBO loss:

$$\max_{\theta,\phi,\psi} \mathcal{L} = \underbrace{\mathbb{E}_{\mathbf{A}\sim\mathbb{P}_{\phi,\psi}(\mathbf{A})}[\log \mathbb{P}_\theta(\mathbf{X}|\mathbf{A})]}_{(i)} - \lambda \underbrace{\text{KL}(\mathbb{P}_{\phi,\psi}(\mathbf{A}) || \mathbb{P}_{\text{prior}}(\mathbf{A}))}_{(ii)} \tag{5}$$

where $\lambda \in \mathbb{R}^+$ is a regularization factor and $\theta, \phi, \psi$ are the parameters of the model to optimize. Indeed, similarly to Eq. 4, both terms (i) and (ii) can be formulated as an expectation over the DAG probabilistic model $\mathbb{P}_{\phi,\psi}(\mathbf{A})$. Importantly, the optimum of the variational inference problem in Eq. 5 is reached when the learned probability distribution over DAGs is equal to the posterior probability distribution over DAGs i.e. $\mathbb{P}(\mathbf{A}|\phi,\psi) \approx \mathbb{P}(\mathbf{A}|\mathcal{D})$ where $\mathcal{D}$ denotes the dataset of observations.

**Variational inference with DP-DAG.** We propose VI-DP-DAG, a new method combining DP-DAG and variational inference to learn the matrix form of a SEM from observational data. At training time, VI-DP-DAG consists of three steps: **(1)** It *differentiably* samples a *valid* DAG adjacency matrix $\mathbf{A}$ from a probabilistic model over DAGs $\mathbb{P}_{\phi,\psi}(\mathbf{A})$. In practice, we parametrize $\mathbb{P}_{\phi,\psi}(\mathbf{A})$ with DP-DAG. **(2)** It applies the $n$ transformations $f_{i,\theta}$ on the observations $\mathbf{X}$ masked with the sampled DAG $\mathbf{A}$ i.e. $\hat{X}_i = f_{i,\theta}(\mathbf{A}_i \circ \mathbf{X}, \epsilon_i)$. In practice, we parametrize $f_{i,\theta}$ with neural networks. **(3)** It jointly updates all parameters $\theta, \phi, \psi$ by maximizing at each iteration an approximation of the ELBO loss in Eq. 5. In practice, we approximate the term (i) by sampling a single DAG matrix $\mathbf{A}$ at each iteration and assume a Gaussian distribution with unit variance around $\hat{\mathbf{X}}$ (i.e. (i) $= ||\mathbf{X} - \hat{\mathbf{X}}||_2$). We compute the term (ii) by setting a small prior $\mathbb{P}_{\text{prior}}(U_{ij})$ on the edge probability (i.e. (ii)$= \sum_{ij} \text{KL}(\mathbb{P}_\phi(U_{ij} || \mathbb{P}_{\text{prior}}(U_{ij})))$ thus acting as a natural sparsity regularization. We set no prior on the permutation probability for two reasons: In theory, a permutation prior is likely biased toward graphs that are compatible with a larger number of orderings (Kuipers & Moffa, 2017; Viinikka et al., 2020). In practice, the closed-form computation of the permutation probability $\mathbb{P}_\theta(\mathbf{\Pi})$ is generally intractable (Mena et al., 2018). Thus, VI-DP-DAG approximates the true posterior distribution over DAG edges probability only, which achieves excellent predictive performance in practice. Beyond variational inference, VI-DP-DAG is theoretically motivated from a second perspective. Assuming that the data comes from a SEM with additive noise, Ng et al. (2019) showed that minimizing the term (i) enforces the sampled matrix $\mathbf{A}$ to represent a super-graph of the ground-truth DAG adjacency matrix $\mathbf{A}^*$ (i.e. all the edges in $\mathbf{A}^*$ are also in $\mathbf{A}$) thus suggesting to add a sparsity regularization term to remove spurious edges. Interestingly, the term (ii) – which arises naturally from the variational inference framework – indeed acts a sparsity regularizer and pushes the probability $\mathbb{P}_\phi(U_{ij})$ to the fixed low prior value $\mathbb{P}_{\text{prior}}(U_{ij})$.

At inference time, we can use VI-DP-DAG in two ways. On one hand, we can extract the cause/consequence relationships from, e.g., one sampled DAG adjacency matrix $A$. On the other hand, we can estimate the missing values $x_i$ from its learned parents and causal mechanisms i.e. $x_i \approx \hat{x}_i = f_{i,\theta}(x_{\text{pa}}(i))$. Intuitively, a small prediction error $||x_i - \hat{x}_i||_2$ indicates that the learned parents $x_{\text{pa}}(i)$ forecast well the child node $x_i$ i.e. the parent nodes $x_{\text{pa}}(i)$ are *Granger-causes* of the child node $x_i$.

## 5 EXPERIMENTS

In this section, we compare VI-DP-DAG to existing methods for differentiable DAG learning on extensive experiments including undirected and directed link prediction, consequence prediction from learned causal mechanisms, and training time. This set-up aims at highlighting the speed and quality of VI-DP-DAG for DAG structure learning and causal mechanisms learning. Further, we evaluate the speed and the optimization quality of the probabilistic model DP-DAG alone.

### 5.1 SET-UP

In our experiments, VI-DP-DAG parametrizes the permutation probability $\mathbb{P}_\psi(\mathbf{\Pi})$ with Gumbel-Sinkhorn or Gumbel-Top-k trick, the edge probability $\mathbb{P}_\phi(\mathbf{U})$ with Gumbel-Softmax distribution and the causal mechanisms $f_{i,\theta}$ with a 3 layers Multi-Layer Perceptron (MLP). We use early stopping and perform a grid search over the permutation probability parametrization (i.e. Gumbel-Sinkhorn or Gumbel-Top-k), the fixed prior probability $\mathbb{P}_{\text{prior}}(U_{ij}) \in [1e^{-2}, 1e^{-1}]$ and the regularization factor $\lambda \in [0, 1e^{-1}]$. Finally, all temperature parameters are fixed to $\tau = 1$ in all experiments.

**Baselines.** We focus our evaluation on *differentiable* methods for DAG learning and compared to **DAG-GNN** (Yu et al., 2019), **DAG-NF** (Wehenkel & Louppe, 2021), **GraN-DAG** (Lachapelle et al., 2020) and **Masked-DAG** (Ng et al., 2019) by using their official implementation when available. Unless otherwise stated, we did not use non-differentiable pre- or post-processing steps for any of the models. For the sake of completeness, we also compare DP-DAG with models using non-differentiable pre- and post-processing like Preliminary Neighbor Selection (PNS) and CAM pruning (Bühlmann et al., 2014) in App. F.3. Similarly to VI-DP-DAG, we perform a grid-search for the hyper-parameters of all models (including learning rate, number of hidden units per linear layer, or regularization factors). For all metrics, we report the mean and the standard error of the mean of the results over 10 runs. We give further details on the models in App. E.

**Datasets.** We use a total of 11 synthetic and real datasets. For the synthetic datasets, we closely follow (Lachapelle et al., 2020; Ng et al., 2019; Peters et al., 2014). The graph structures $\mathbf{A}^*$ were generated with the Erdös-Rényi (ER) or Scale-Free (SF) network models, and the causal mechanisms $f_i^*$ were generated from a (non-linear) Gaussian Process with RBF kernel of bandwidth one and with independent zero-mean Gaussian noises $\epsilon_i$. We considered different sizes of graphs including number of nodes $n$ in $\{10, 50, 100\}$ and number of edges $m$ in $\{10, 40, 50, 100, 200, 400\}$. We denote the synthetic datasets by **SF/ER-$n$-$m$**. We used 10 sampled datasets per setting. For the real datasets, we closely follow (Koller & Friedman, 2009; Lachapelle et al., 2020; Ng et al., 2019; Yu et al., 2021; Zheng et al., 2018). We use the **Sachs** dataset which measures the expression level of different proteins and phospholipids in human cells (Sachs et al., 2005). We also use the pseudo-real **SynTReN** dataset sampled from a generator that was designed to create synthetic transcriptional regulatory networks and produces simulated gene expression data that approximates experimental data (Van den Bulcke et al., 2006). For all these datasets, we can access the ground-truth DAG adjacency matrices $\mathbf{A}^*$ used to generate the data or estimated by experts. We split all datasets in training/validation/test sets with $80\%/10\%/10\%$. We give further details on the datasets in Sec. D in the appendix.

### 5.2 RESULTS

**DAG sampling.** We show results for the DAG sampling time in Fig. 3 and Fig. 4 in the appendix. While both Gumbel Sinkhorn and Gumbel-Top-k parametrizations can sample a DAG with 200 nodes in less than 1 second, Gumbel-Top-k is significantly faster than Gumbel-Sinkhorn as expected from the complexity analysis. Further, we show the results of differentiable sampling optimization using DP-DAG for DAG learning when the ground-truth DAG is observed in Tab. 6 in the appendix. On this toy task, DP-DAG recovers almost perfectly the ground-truth DAG structure. These observations validate that DP-DAG is a reliable method for *fast* and *differentiable* DAG sampling. Further details about these two experiments are in App. F.1.

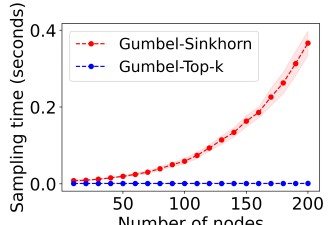

Figure 3: Sampling time of DP-DAG using Gumbel-Sinkhorn or Gumbel-Top-k.

**DAG structure.** We evaluate the learning of the DAG structure by comparing the ground-truth adjacency matrix $A_{ij}^*$ to the learned score $S_{ij}$. For baselines, we use entries from the weighted adjacency matrix $S_{ij} = A_{ij}$ as scores for directed edges, and $S_{ij} = A_{ij} + A_{ji}$ as scores for undirected edges. For DP-DAG, we use the edge probability $S_{ij} = \mathbb{P}_\phi(U_{\pi(i)\pi(j)})$ as scores for directed edges and $S_{ij} = \mathbb{P}_\phi(U_{\pi(i)\pi(j)}) + \mathbb{P}_\phi(U_{\pi(j)\pi(i)})$ as scores for undirected edges. In this case, the permutation $\pi$ (or equivalently $\mathbf{\Pi}$) is deterministically computed by removing Gumbel noise in the forward sampling step. The directed scores (**Dir-**) indicate if an edge *with a specific direction* exists between two nodes. The undirected scores (**Un-**) indicates if an edge *without a specific direction* exists between two nodes. We compare the directed and undirected scores to the ground-truth binary adjacency matrix by using the area under the curve of precision-recall (**AUC-PR**) and the area under the receiver operating characteristic curve (**AUC-ROC**). These metrics have the important advantage to be independent of a threshold choice (Vowels et al., 2021). Thus, AUC-PR and AUC-ROC are better indicators of the true performance of the models contrary to other structure metrics like Structural Hamming Distance which needs to arbitrary select a threshold.

We show the results for *differentiable* DAG structure learning on synthetic and real datasets in Tab. 1 and in Tab. 7 in the appendix. In total, VI-DP-DAG achieves 27/44 best scores and second-best scores otherwise. On synthetic datasets, only GraN-DAG competes with VI-DP-DAG on small datasets with 10 nodes. DP-DAG outperforms all baselines on all larger synthetic datasets with 50 or 100 nodes for both ER and SF graph types. E.g. on the ER-100-400 dataset, VI-DP-DAG gives an absolute improvement of $+17\%$ for Un-AUC-PR, $+25\%$ for Un-AUC-ROC, $+13\%$ for Dir-AUC-PR and $+9\%$ for Dir-AUC-ROC. We observe that DAG-GNN and DAG-NF perform comparably to random guessing corresponding to an AUC-ROC of $50\%$ on those datasets. We hypothesize that their poor performances might be due to their incorrect modeling assumptions or their incapacity to model this type of causal relationships as mentioned in Lachapelle et al. (2020); Yu et al. (2019). On real datasets, no baseline consistently outperforms VI-DP-DAG which obtains 4/8 best scores and second-best scores otherwise. We observe on both synthetic and real datasets that VI-DP-DAG performs better with undirected edges scores than with directed edges scores indicating that VI-DP-DAG might invert the edge directions. This can be intuitively explained by the presence of node pairs where $x_i$ is an equally good predictor of $x_j$ as $x_j$ is a good predictor of $x_i$. Finally, we show further evidence that VI-DP-DAG learns reliable DAG structures under perturbations in App. F.5. Indeed it confidently assigns lower confidence scores to perturbed versions of the ground-truth DAG.

We show the results for DAG structure learning with *non-differentiable* pre- and post-processing in Tab. 8 and Tab. 9 in the appendix. In these experiments, we apply the default pre-processing PNS and post-processing CAM pruning steps to CAM (Bühlmann et al., 2014), GraN-DAG (Lachapelle et al., 2020), Masked-DAG (Ng et al., 2019) as suggested by the original papers. We similarly apply CAM pruning to the final results of VI-DP-DAG. In this case, we observe that the greedy CAM is the best performing model for most synthetic datasets while VI-DP-DAG maintains competitive performance. Interestingly, VI-DP-DAG with non-differentiable pre- and post-processing is the best performing model on real datasets (i.e. 7/8 best scores). Further, we notice that the pre- and post-processing steps raise significantly the scores of GraN-DAG and Masked-DAG which becomes more competitive with VI-DP-DAG, while VI-DP-DAG benefits from it to a lower extent (e.g. see ER-100-400 in Tab. 1 and Tab. 8). This suggests that methods based on augmented Lagrangian optimization are more dependent on additional non-differentiable processing steps than VI-DP-DAG. Lachapelle et al. (2020) make similar observations on the dependence on PNS and CAM pruning steps.

**Causal mechanisms.** We evaluate the causal mechanisms learning by computing the Mean Squared Error (**MSE**) between the ground-truth node values $x_i$ and the estimated node values $\hat{x}_i = f_{i,\theta}(x_{\mathrm{pa}}(i))$ on a new test set. It can be computed in a vectorized way i.e. $||\mathbf{X} - \hat{\mathbf{X}}||_2^2$ where $\hat{X}_i = f_{i,\theta}(\mathbf{A}_i \circ \mathbf{X})$. This task is particularly important when predicting missing values from the values of known parent causes. For this experiment, we compare to other baselines which jointly learn the adjacency matrix $\mathbf{A}$ and the causal mechanisms $f_{i,\theta}$ explicitly. Hence, we compare to Masked-DAG and GraN-DAG. At testing time, we remove the remaining cycles from the adjacency matrix $\mathbf{A}$ learned with augmented Lagrangian optimization to guarantee a valid DAG structure as suggested by Lachapelle et al. (2020); Ng et al. (2019). Further, we also compare to **GT-DAG** which uses the ground-truth DAG adjacency matrix $\mathbf{A}^*$ at training and testing time. GT-DAG can be considered as a lower bound for the MSE scores. It indicates if other methods manage to jointly learn well both the DAG structure $\mathbf{A}$ and the causal mechanisms $f_{i,\theta}$. For DP-DAG, the adjacency matrix $\mathbf{A}$ is computed by deterministically computing $\pi$ (or equivalently $\mathbf{\Pi}$) without Gumbel noise in the forward sampling step, and second

| Models | Un-AUC-PR | Un-AUC-ROC | Dir-AUC-PR | Dir-AUC-ROC |
|---|---|---|---|---|
| **DAG-GNN** | $18.67 \pm 0.17$ | $48.44 \pm 0.15$ | $8.21 \pm 0.15$ | $47.83 \pm 0.34$ |
| **DAG-NF** | $16.11 \pm 0.12$ | $50.20 \pm 0.03$ | $8.15 \pm 0.07$ | $50.19 \pm 0.02$ |
| **GraN-DAG** | $\underline{25.11 \pm 0.4}$ | $\underline{56.57 \pm 0.35}$ | $\underline{17.61 \pm 0.47}$ | $56.35 \pm 0.34$ |
| **Masked-DAG** | $17.55 \pm 0.14$ | $51.09 \pm 0.05$ | $9.14 \pm 0.09$ | $50.81 \pm 0.04$ |
| **VI-DP-DAG** | $\mathbf{62.08} \pm 0.34$ | $\mathbf{84.53} \pm 0.25$ | $\mathbf{41.70} \pm 0.39$ | $\mathbf{71.08} \pm 0.33$ |

(a) ER-50-200

| Models | Un-AUC-PR | Un-AUC-ROC | Dir-AUC-PR | Dir-AUC-ROC |
|---|---|---|---|---|
| **DAG-GNN** | $\underline{14.65 \pm 0.25}$ | $49.05 \pm 0.2$ | $\underline{5.56 \pm 0.09}$ | $47.47 \pm 0.29$ |
| **DAG-NF** | $7.82 \pm 0.04$ | $49.98 \pm 0.02$ | $3.94 \pm 0.02$ | $50.05 \pm 0.02$ |
| **GraN-DAG** | $9.73 \pm 0.11$ | $\underline{52.82 \pm 0.11}$ | $4.42 \pm 0.04$ | $\underline{51.04 \pm 0.06}$ |
| **Masked-DAG** | $8.93 \pm 0.12$ | $50.65 \pm 0.06$ | $4.73 \pm 0.1$ | $50.52 \pm 0.06$ |
| **VI-DP-DAG** | $\mathbf{36.98} \pm 0.98$ | $\mathbf{77.23} \pm 0.63$ | $\mathbf{19.00} \pm 1.07$ | $\mathbf{60.35} \pm 1.01$ |

(b) ER-100-400

| Models | Un-AUC-PR | Un-AUC-ROC | Dir-AUC-PR | Dir-AUC-ROC |
|---|---|---|---|---|
| **DAG-GNN** | $14.94 \pm 0.17$ | $51.41 \pm 0.26$ | $6.91 \pm 0.12$ | $50.31 \pm 0.27$ |
| **DAG-NF** | $11.83 \pm 0.1$ | $50.18 \pm 0.03$ | $5.86 \pm 0.06$ | $50.02 \pm 0.02$ |
| **GraN-DAG** | $\underline{18.75 \pm 0.29}$ | $\underline{54.76 \pm 0.17}$ | $\underline{12.35 \pm 0.32}$ | $\underline{53.31 \pm 0.43}$ |
| **Masked-DAG** | $14.39 \pm 0.13$ | $51.64 \pm 0.06$ | $7.23 \pm 0.12$ | $50.97 \pm 0.06$ |
| **VI-DP-DAG** | $\mathbf{33.81} \pm 0.39$ | $\mathbf{68.18} \pm 0.39$ | $\mathbf{17.46} \pm 0.34$ | $\mathbf{59.70} \pm 0.65$ |

(c) SF-50-200

| Models | Un-AUC-PR | Un-AUC-ROC | Dir-AUC-PR | Dir-AUC-ROC |
|---|---|---|---|---|
| **DAG-GNN** | $\mathbf{49.92} \pm 0.46$ | $\underline{59.59 \pm 0.57}$ | $\mathbf{23.84} \pm 0.52$ | $51.67 \pm 0.62$ |
| **DAG-NF** | $35.66 \pm 0.32$ | $53.57 \pm 0.44$ | $15.43 \pm 0.11$ | $50.16 \pm 0.31$ |
| **GraN-DAG** | $34.28 \pm 0.32$ | $54.57 \pm 0.36$ | $19.95 \pm 0.35$ | $\underline{55.91 \pm 0.33}$ |
| **Masked-DAG** | $36.98 \pm 0.25$ | $55.13 \pm 0.15$ | $17.74 \pm 0.31$ | $51.71 \pm 0.23$ |
| **VI-DP-DAG** | $\underline{43.37 \pm 0.55}$ | $\mathbf{59.78} \pm 0.62$ | $\underline{22.96 \pm 0.81}$ | $\mathbf{60.02} \pm 0.91$ |

(d) Sachs

| Models | Un-AUC-PR | Un-AUC-ROC | Dir-AUC-PR | Dir-AUC-ROC |
|---|---|---|---|---|
| **DAG-GNN** | $\underline{24.83 \pm 0.19}$ | $\underline{66.75 \pm 0.2}$ | $9.26 \pm 0.11$ | $59.61 \pm 0.19$ |
| **DAG-NF** | $12.89 \pm 0.15$ | $49.02 \pm 0.24$ | $6.63 \pm 0.15$ | $49.11 \pm 0.16$ |
| **GraN-DAG** | $19.89 \pm 0.4$ | $62.29 \pm 0.31$ | $\mathbf{15.39} \pm 0.44$ | $\mathbf{63.91} \pm 0.34$ |
| **Masked-DAG** | $16.66 \pm 0.38$ | $53.93 \pm 0.28$ | $7.99 \pm 0.29$ | $52.26 \pm 0.35$ |
| **VI-DP-DAG** | $\mathbf{26.33} \pm 0.56$ | $\mathbf{70.84} \pm 0.45$ | $\underline{14.17 \pm 0.76}$ | $\underline{59.84 \pm 1.46}$ |

(e) SynTReN

Table 1: DAG structure learning results on synthetic and real datasets for all differentiable DAG learning models with AUC-PR and AUC-ROC scores (Higher is better). *Un-* and *Dir-* indicate scores for undirected and directed edges. Best scores among all models are in bold. Second best scores among all models are underlined.

keeping the edges such that $\mathbb{P}_\phi(U_{\pi(i)\pi(j)}) > .5$. These edges intuitively correspond to parent relationships which are more likely to be present than absent according to the model.

We show the results for causal mechanisms learning in Tab. 2. Overall, DP-DAG is the most competitive model and achieves 9/11 best scores. Similarly to DAG structure learning, GraN-DAG is competitive for small graphs with 10 nodes but VI-DP-DAG outperforms all baselines for most larger graphs with 50 and 100 nodes. Interestingly, DP-DAG brings a particularly significant improvement on real datasets compared to GraN-DAG and Masked-DAG with scores very close to GT-DAG. This suggests that the potentially noisy ground-truth DAGs estimated by experts on these real datasets have an equivalent predictive power as the DAG learned by VI-DP-DAG. This aligns well with the intuition that VI-DP-DAG learns Granger causal relationships. Finally, we evaluate in App. F.6 the impact of the threshold choice $t$ on the edge probabilities $\mathbb{P}_\phi(U_{\pi(i)\pi(j)}) > t$. We observe that the MSE score of DP-DAG achieves stable high performance regardless of the threshold value.

**Training time.** We evaluate the training time of all models on a single GPU (NVIDIA GTX 1080 Ti, 11 GB memory). Training time is particularly relevant for DAG structure learning since predicting the adjacency matrix $\mathbf{A}$ from a new dataset represents a full training. Thus, real-world applications of DAG learning generally aim at fast training time.

We show the results in Tab. 3. The training of VI-DP-DAG is one order of magnitude faster than all models on all datasets. Hence, assuming that training time is typically linearly correlated with compute cost, energy consumption and $CO_2$ emission, the fast training DP-DAG significantly improves the applicability of DAG learning methods. In particular, VI-DP-DAG is from $\times 5$ to $\times 18$ faster to train than GraN-DAG which is the second-best performing model for DAG structure learning and causal mechanisms learning. The training speed difference can be explained by the different optimization schemes. Indeed, Lagrangian optimization typically requires solving $T \approx 10$ successive optimization problems (Zheng et al., 2018) while VI-DP-DAG only requires solving a single optimization problem. Finally, by aggregating training and sampling time, VI-DP-DAG requires around 191 seconds to sample a DAG with 100 nodes from the learned approximate posterior distribution. In contrast, other sampling methods based on MCMC sampling require running the MCMC methods for 12 hours to sample a DAG with 100 nodes (Viinikka et al., 2020).

| | GraN-DAG | Masked-DAG | VI-DP-DAG | GT-DAG* |
|---|---|---|---|---|
| **ER-10-10** | $\mathbf{0.61} \pm 0.01$ | $0.93 \pm 0.02$ | $0.69 \pm 0.01$ | $0.58 \pm 0.01$ |
| **ER-10-40** | $\mathbf{0.41} \pm 0.01$ | $1.0 \pm 0.02$ | $0.49 \pm 0.01$ | $0.3 \pm 0.0$ |
| **ER-50-50** | $0.82 \pm 0.01$ | $0.97 \pm 0.0$ | $\mathbf{0.8} \pm 0.01$ | $0.57 \pm 0.01$ |
| **ER-50-200** | $0.84 \pm 0.01$ | $0.97 \pm 0.0$ | $\mathbf{0.8} \pm 0.01$ | $0.48 \pm 0.01$ |
| **ER-100-100** | $0.96 \pm 0.02$ | $0.95 \pm 0.0$ | $\mathbf{0.84} \pm 0.0$ | $0.54 \pm 0.0$ |
| **ER-100-400** | $0.96 \pm 0.02$ | $0.98 \pm 0.0$ | $\mathbf{0.9} \pm 0.0$ | $0.48 \pm 0.0$ |
| **SF-10-10** | $\mathbf{0.74} \pm 0.01$ | $0.95 \pm 0.02$ | $\mathbf{0.74} \pm 0.01$ | $0.66 \pm 0.01$ |
| **SF-50-50** | $\mathbf{0.93} \pm 0.01$ | $0.99 \pm 0.0$ | $\mathbf{0.93} \pm 0.0$ | $0.79 \pm 0.01$ |
| **SF-50-200** | $\mathbf{0.9} \pm 0.01$ | $0.95 \pm 0.0$ | $\mathbf{0.9} \pm 0.0$ | $0.77 \pm 0.0$ |
| **Sachs** | $1.16 \pm 0.05$ | $0.93 \pm 0.05$ | $\mathbf{0.86} \pm 0.04$ | $0.84 \pm 0.04$ |
| **SynTReN** | $0.82 \pm 0.02$ | $0.8 \pm 0.02$ | $\mathbf{0.21} \pm 0.0$ | $0.21 \pm 0.0$ |

Table 2: Causal mechanisms learning results on all datasets with MSE score (Lower is better) on a test set i.e. $||\mathbf{X} - \hat{\mathbf{X}}||_2^2$ where $\hat{X}_i = \hat{f}_{i,\theta}(\mathbf{A}_i \circ \mathbf{X})$. Best scores among all models which jointly learn the adjacency matrix $\mathbf{A}$ and the causal mechanisms $f_\psi$ are in bold. GT-DAG* is excluded since it is the ideal scenario when the ground-truth $\mathbf{A}^*$ is known at training and testing time.

| | DAG-GNN | GraN-DAG | Masked-DAG | DAG-NF | VI-DP-DAG |
|---|---|---|---|---|---|
| **ER-10-10** | $2997 \pm 20$ | $596 \pm 8$ | $1997 \pm 30$ | $5191 \pm 53$ | $\mathbf{160} \pm 6$ |
| **ER-10-40** | $10727 \pm 145$ | $615 \pm 12$ | $1919 \pm 26$ | $4962 \pm 44$ | $\mathbf{86} \pm 3$ |
| **ER-50-200** | $3485 \pm 110$ | $1290 \pm 19$ | $4279 \pm 73$ | $5329 \pm 63$ | $\mathbf{190} \pm 4$ |
| **ER-50-50** | $3260 \pm 72$ | $1078 \pm 22$ | $5121 \pm 83$ | $5531 \pm 71$ | $\mathbf{194} \pm 3$ |
| **ER-100-100** | $4077 \pm 50$ | $2503 \pm 54$ | $10530 \pm 212$ | $4969 \pm 38$ | $\mathbf{138} \pm 3$ |
| **ER-100-400** | $5198 \pm 66$ | $2391 \pm 59$ | $10629 \pm 200$ | $5008 \pm 36$ | $\mathbf{191} \pm 7$ |
| **SF-10-10** | $1642 \pm 17$ | $641 \pm 24$ | $1873 \pm 29$ | $4887 \pm 31$ | $\mathbf{83} \pm 2$ |
| **SF-50-200** | $2969 \pm 32$ | $1138 \pm 44$ | $4611 \pm 58$ | $4901 \pm 23$ | $\mathbf{112} \pm 2$ |
| **SF-50-50** | $2910 \pm 20$ | $1087 \pm 40$ | $4640 \pm 60$ | $4888 \pm 23$ | $\mathbf{87} \pm 1$ |
| **Sachs** | $1570 \pm 66$ | $358 \pm 13$ | $1657 \pm 36$ | $4970 \pm 26$ | $\mathbf{67} \pm 3$ |
| **SynTReN** | $8257 \pm 160$ | $461 \pm 9$ | $5120 \pm 129$ | $4926 \pm 25$ | $\mathbf{53} \pm 1$ |

Table 3: Training time for all differentiable DAG learning models in seconds (Lower is better). Fastest training time among all models are in bold.

## 6 CONCLUSION

We propose a Differentiable Probabilistic model over DAGs (DP-DAG) which allows *fast* and *differentiable* DAG sampling and can be implemented in few lines of code. To this end, DP-DAG uses differentiable permutation and edges sampling based on Gumbel-Sinkhorn, Gumbel-Top-K and Gumbel-Softmax tricks. We propose VI-DP-DAG, a new method combining variational inference and DP-DAG for DAG learning from observational data with continuous optimization. VI-DP-DAG guarantees valid DAG prediction at *any* time during training. In our extensive experiments, VI-DP-DAG performs favorably to other differentiable DAG learning baselines for DAG structure and causal mechanisms learning on synthetic and real datasets while training one order of magnitude faster.

## 7   ETHICS STATEMENT

The Assessment List for Trustworthy AI (ALTAI) (Com., 2020) includes reliability, transparency and Environmental well-being. Accurate and fast causal discovery from observational data is important to give meaningful and interpretable predictions at a low energy cost. Therefore, VI-DP-DAG brings a significant improvement regarding these values by performing high-quality DAG structure and causal mechanisms predictions at a low training cost.

While DP-DAG achieves high-performance for DAG structure learning and causal mechanisms learning, there is always a non-negligible risk that DP-DAG does not fully capture the real-world complexity. Thus, we raise awareness about the risk of excessive trust in causal predictions from Machine learning models. This is crucial when causal predictions are applied in domains subject to privacy or fairness issues (e.g. finance, medicine, policy decision making, etc). In particular, VI-DP-DAG does not provide any guarantee on the causal interpretation of the edges in the final predicted DAG. Therefore, we encourage practitioners to proactively confront the model predictions for DAG structure and causal mechanisms to desired behaviors in real-world use cases.

## 8   REPRODUCIBILITY STATEMENT

We provide all datasets and the model code at the project page [1]. In App. D in the appendix, we give a detailed description for each dataset used in this paper. This description includes the task description, the dataset size, the number of nodes and edges and the train/validation/test splits used in the experiments. We further explain the hardware used for the experiments. In App. E in the appendix, we give a detailed description of the architecture and grid search performed for each model used in this paper. We describe the metrics used in the experiments in App. 5.

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

# A    AUGMENTED LAGRANGIAN OPTIMIZATION

In this section we recall the augmented Lagrangian optimization used by (Lachapelle et al., 2020; Ng et al., 2019; Wehenkel & Louppe, 2021; Yu et al., 2019; Zheng et al., 2018). To this end, we denote the learned adjacency matrix $\mathbf{A}$ and the other model parameters by $\theta$. Hence, we can rewrite the augmented Lagrangian optimization problem as follows:

$$\min_G \ score(\mathbf{X}, \mathbf{A}, \theta) \tag{6}$$

$$\text{s.t. } h(\mathbf{A}) = 0 \tag{7}$$

where $h(\mathbf{A}) = \mathrm{Tr}(e^{\mathbf{A}}) - n$ or $h(\mathbf{A}) = \mathrm{Tr}(\mathbf{I} + \alpha \mathbf{A} \circ \mathbf{A})^n - n$ with $\alpha \in \mathbb{R}^+$. The function $h$ is a smooth function over weighted adjacency matrices $\mathbf{A} \in \mathbb{R}^{n \times n}$ which is equal to zero when the corresponding graph $G$ satisfies the DAG constraints. The augmented Lagrangian of the problem 6 is defined as:

$$L_c(\mathbf{A}, \theta, \lambda) = score(\mathbf{X}, \mathbf{A}, \theta) + \lambda h(\mathbf{A}) + \frac{c}{2}|h(\mathbf{A})|^2$$

where $\lambda$ is the Lagrangian multiplier and $c$ is the penalty parameter. During training, a practical optimization scheme is to update the parameters $\lambda$ and $c$ with the iteration:

$$\mathbf{A}_k, \theta_k = \arg\min_{\mathbf{A}, \theta} L_c(\mathbf{A}, \theta, \lambda)$$

$$\lambda_{k+1} = \lambda_k + c_k h(\mathbf{A}_k)$$

$$c_{k+1} = \begin{cases} \eta c_k, & \text{if } |h(\mathbf{A}_k)| > \gamma |h(\mathbf{A}_{k-1})| \\ c_k, & \text{otherwise} \end{cases}$$

where $\eta > 1$ and $\gamma > 1$ are hyper-parameters. The augmented Lagrangian optimization procedure is expensive since each Lagrangian iteration requires to solve a minimization problem which can be done with stochastic optimization methods. In practice, previous works set that $\eta = 10$ and $\gamma = \frac{1}{4}$ and use a stopping criterion $h(\mathbf{A}_k) < \epsilon \in \{1e^{-6}, 1e^{-8}, 1e^{-10}\}$ to decide for the end of optimization where $\epsilon$ is a constraint tolerance (Lachapelle et al., 2020; Ng et al., 2019; Yu et al., 2019; Zheng et al., 2018). This generally leads to up to 10 Lagrangian iterations (Zheng et al., 2018) and does not guarantee acyclicity of the final adjacency matrix (Lachapelle et al., 2020; Ng et al., 2019; Zheng et al., 2018). Hence, the augmented Lagrangian requires a final non-differentiable thresholding step to remove remaining cycles. In contrast, DP-DAG can always output valid sampled DAG solutions.

# B    GUMBEL-SOFTMAX, GUMBEL-SINKHORN, GUMBEL-TOP-K DISTRIBUTIONS

## B.1    GUMBEL-SOFTMAX DISTRIBUTION (JANG ET AL., 2017)

We assume that a variable $\mathbf{U}$ on the $(k-1)$-dimensional simplex follows a Gumbel-Softmax distribution with class probabilities $\phi_1, ..., \phi_k$ and temperature parameter $\tau$ i.e.:

$$U \sim \text{Gumbel-Softmax}_\tau(\phi)$$

The Gumbel-Softmax distribution becomes identical to the categorical distribution when the temperature $\tau$ approaches 0. The density of the Gumbel-Softmax can be expressed as follows:

$$\mathbb{P}(\mathbf{U}|\phi, \tau) = \Gamma(k)\tau^{k-1}\frac{\prod_{i=1}^k \phi_i/U_i^{\tau+1}}{\sum_{i=1}^k \phi_i/U_i^{\tau+1}}$$

where $\Gamma(k)$ denotes the Gamma function. We can obtain a sample $\mathbf{U}$ from the Gumbel-Softmax distribution by computing:

$$U_i = \frac{e^{(\log(\phi_i) + G_i)/\tau}}{\sum_{j=1}^k e^{(\log(\phi_j) + G_j)/\tau}}$$

whre $G_i$ are i.i.d. samples from a Gumbel$(0)$ distribution. For differentiable sampling, we can use the straight-through estimator (Bengio et al., 2013). Hence, the discrete variable $\text{one\_hot}(\arg\max(\mathbf{U}))$ is used for the forward pass, and the continuous variable $\mathbf{U}$ is used for the backward pass.

## B.2 Gumbel-Sinkhorn distribution (Mena et al., 2018)

We assume that a variable $\mathbf{\Pi}$ on the space of permutation matrix follows the Gumbel-Sinkhorn distribution with parameters $\psi \in \mathbb{R}^{n \times n}$ and temperature parameter $\tau$ i.e.:

$$\mathbf{\Pi} \sim \text{Gumbel-Sinkhorn}_\tau(\psi)$$

The Gumbel-Sinkhorn distribution becomes identical to the Gumbel-Matching distribution when the temperature $\tau$ approaches 0. Neither the Gumbel-Matching or the Gumbel-Sinkhorn distributions have tractable densities. We can obtain a sample $\mathbf{\Pi}$ from the Gumbel-Sinkhorn distribution by computing:

$$\mathbf{\Pi} = S(\psi + \mathbf{G})$$

where $S$ is the Sinkhorn operator which iteratively normalizes rows and columns of a matrix and $\mathbf{G} \in \mathbb{R}^{n \times n}$ is a matrix of standard i.i.d. Gumbel noise. For differentiable sampling, we can use the straight-through estimator (Bengio et al., 2013). Hence, the discrete variable $\mathbf{\Pi} = \text{Hungarian}(\hat{\mathbf{\Pi}})$ after applying the Hungarian algorithm (Munkres, 1957) to get a discrete permutation is used for the forward pass, and the continuous variable $\mathbf{\Pi}$ is used for the backward pass. Finally, we can deterministically compute a permutation by removing all Gumbel noise in the forward pass.

## B.3 Gumbel-Top-K distribution (Kool et al., 2019)

We assume that the variable $\pi$ on the space of ordered sample of size $k$ without replacement follows the Gumbel-Top-k distribution with parameters $\psi \in \mathbb{R}^n$ i.e.:

$$\pi \sim \text{Gumbel-Top-k}_\tau(\psi)$$

where $\tau$ is an optional temperature parameter usually set to 1. The density of the Gumbel-Top-k can be expressed as follows:

$$\mathbb{P}(\pi|\psi, \tau) = \prod_{i=1}^{k} \frac{e^{\psi_{\pi_j}}}{\sum_{l \in N_j} e^{\psi_l}}$$

where $N_j = \{1, ..., n\} \backslash \{\pi_1, ..., \pi_{j-1}\}$. This corresponds to subsequently sampling $k$ times without replacement from the categorical distribution over the nodes parametrized with $\psi$ with renormalization after each step. We can compute a sample $\pi$ from the Gumbel-Top-k distribution by computing:

$$\pi = \arg \text{top-k}(\psi + \mathbf{G})$$

where $\arg \text{top-k}$ is the operator selecting the indices with the top-k largest values and $\mathbf{G} \in \mathbb{R}^n$ is a vector of i.i.d. Gumbel noise. Importantly, remark that the Gumbel-Top-n distribution (i.e. $k = n$) states the sorted perturbed log-probabilities $\pi = \text{Sort}(\psi + \mathbf{G})$ where $\mathbf{G} \in \mathbb{R}^n \sim \text{Gumbel}(0)$ defines a distribution over component-wise permutation (or linear ordering without replacement). Finally, we can deterministically compute a permutation by removing all Gumbel noise in the forward pass.

## C SoftSort Operator (Prillo & Eisenschlos, 2020)

The SoftSort operator with temperature parameter $\tau$ is defined as:

$$\text{SoftSort}_\tau^d(s) = \text{Softmax}\left(\frac{-d(\text{sort}(s)\mathbb{1}^T, \mathbb{1}s^T)}{\tau}\right)$$

where $d$ is any differentiable almost everywhere, semi-metric function (e.g. $d = |\cdot|$). It defines a family as simple continuous relaxation for the permutation matrix corresponding to the argsort operator $\mathbf{\Pi}_{\arg \text{sort}(s)}$. Indeed, we have:

$$\lim_{\tau \to 0} \text{SoftSort}_\tau^d(s) = \mathbf{\Pi}_{\arg \text{sort}(s)}$$

In particular, this limits holds almost surely if the entries of $s$ are drawn from a distribution that is absolutely continuous w.r.t. the Lebesgue measure on $\mathbb{R}$. Further, note that $\text{SoftSort}_\tau^d(s)$ is a unimodal row stochastic matrix and the permutation $\arg \max(\text{SoftSort}_\tau^d(s))$ is equal to $\arg \text{sort}(s)$.

# D  DATASETS

All the datasets are split in train/validation/test splits with $80\%/10\%/10\%$. We use 10 different seeds for the dataset splits. We make all datasets available at the project page [2].

**Synthetic:** We follow the data generation used in related works (Lachapelle et al., 2020; Ng et al., 2019; Peters et al., 2014). The graph structures $A^*$ are generated with the Erdös-Rényi (ER) or Scale-Free (SF), and the causal mechanisms $f_i^*$ are generated from a (non-linear) Gaussian Process with RBF kernel of bandwidth one and with independent zero-mean Gaussian noises $\epsilon_i$. We considered different sizes of graphs including number of nodes $n$ in $\{10, 50, 100\}$ and number of edges $m$ in $\{10, 40, 50, 100, 200, 400\}$. In our experiments, we use 10 sampled datasets per setting. Each dataset contains 1000 samples.

**Sachs (Sachs et al., 2005):** The Sachs dataset has been widely used in the DAG learning literature (Koller & Friedman, 2009; Lachapelle et al., 2020; Ng et al., 2019; Yu et al., 2021; Zheng et al., 2018). It contains 853 observational data points. Each data point measures the expression level of different proteins and phospholipids in human cells. The ground-truth DAG has 11 nodes and 17 edges.

**SynTReN (Van den Bulcke et al., 2006):** The SynTReN dataset has already been used in (Lachapelle et al., 2020). Ten datasets are generated using the Syntren generator [3] with software default parameters except for th complex 2-regulator interactions probability which was set to 1. Each dataset contains 500 samples. The ground-truth DAG has always 20 nodes.

# E  MODELS

In this section, we provide more details about the training of all the models. We train all models on a single GPU (NVIDIA GTX 1080 Ti, 11 GB memory). All numbers are averaged over 10 seeds for the model initialization. We make available the implementation of VI-DP-DAG at the project page [4].

**VI-DP-DAG (ours):** VI-DP-DAG parametrizes the permutation probability $\mathbb{P}_\psi(\mathbf{\Pi})$ with Gumbel-Sinkhorn or Gumbel-Top-K tricks, the edge probability $\mathbb{P}_\phi(\mathbf{U})$ with Gumbel-Softmax distribution and the causal mechanisms $f_{i,\psi}$ with neural networks with 3 linear layers. We performed a grid search over the learning rate $lr \in [1e^{-4}, 1e^{-2}]$, the number of hidden units per linear layer $h \in \{8, 16, 64\}$, the permutation probability parametrization (i.e. Gumbel-Sinkhorn or Gumbel-Top-K), the fixed prior probability $\mathbb{P}_{\text{prior}}(U_{ij}) \in [1e^{-2}, 1e^{-1}]$ and the regularization factor $\lambda \in [0, 1e^{-1}]$. The temperature parameter for Gumbel-Softmax distribution was fixed to $\tau = 1$ in all experiments. VI-DP-DAG is trained using the Adam optimizer (Kingma & Ba, 2015). We perform early stopping by checking loss improvement on the validation set every two epochs and a patience $p = 10$.

**DAG-GNN (Yu et al., 2019):** DAG-GNN model proposes a Graph Neural Network (GNN) autoencoder using variational inference. The latent variable is *not* the DAG adjacency matrix $A$ but a noise matrix $Z$. DAG-GNN uses the augmented Lagrangian approach for optimization. We follow the recommendation for augmented Lagrangian optimization in (Lachapelle et al., 2020; Ng et al., 2019; Yu et al., 2019; Zheng et al., 2018) and set $\eta = 10$ and $\gamma = \frac{1}{4}$. We used a neural network with 3 linear layers for the encoder and decoder architectures. Further, we performed a grid-search on the learning rate $lr \in [1e^{-4}, 1e^{-2}]$ and the number of hidden units per linear layer $h \in \{8, 16, 64\}$. DAG-GNN is trained using the Adam optimizer (Kingma & Ba, 2015). This model does not directly apply for consequence prediction from learned causal mechanisms as it does not predict the consequence values from the values of the learned parents. We use the implementation provided in `https://github.com/fishmoon1234/DAG-GNN`.

**DAG-NF (Wehenkel & Louppe, 2021):** DAG-NF proposes to combine Normalizing Flows with an augmented Lagrangian optimization to learn DAG structures from observational data. We follow the recommendation for augmented Lagrangian optimization in (Lachapelle et al., 2020; Ng et al., 2019; Yu et al., 2019; Zheng et al., 2018) and set $\eta = 10$ and $\gamma = \frac{1}{4}$. Further, we performed a grid-search on the monotonic or affine normalizer architectures, the conditioner and normalizer

---

[2]`https://www.daml.in.tum.de/differentiable-dag-sampling/`

[3]`bioinformatics.intec.ugent.be/kmarchal/SynTReN/index.html`

[4]`https://www.daml.in.tum.de/differentiable-dag-sampling/`

learning rate $lr \in [1e^{-4}, 1e^{-2}]$, the number of hidden units per linear layer in the conditioner and normalizer architectures $h \in \{50, 100, 150, 200\}$ and different l1 regularization $\lambda_{l_1} \in [0, 60]$. We used a neural network with 3 linear layers for the conditioner and normalizer architectures and the embedding size was set to set to 30 as suggested by the authors Wehenkel & Louppe (2021). DAG-NF is trained using the Adam optimizer (Kingma & Ba, 2015). This model does not directly apply for consequence prediction from learned causal mechanisms as it does not predict the consequence values from the values of the learned parents. We use the official implementation provided in `https://github.com/AWehenkel/Graphical-Normalizing-Flows`.

**Gran-DAG (Lachapelle et al., 2020):** Gran-DAG proposes to compute the adjacency matrix from the learned weight of the non-linear causal relationships. Gran-DAG uses the augmented Lagrangian approach for optimization. We follow the recommendation for augmented Lagrangian optimization in (Lachapelle et al., 2020; Ng et al., 2019; Yu et al., 2019; Zheng et al., 2018) and set $\eta = 10$ and $\gamma = \frac{1}{4}$. We used a neural network with 3 linear layers. Further, we performed a grid-search on the learning rate $lr \in [1e^{-4}, 1e^{-2}]$ and the number of hidden units per linear layer $h \in \{8, 16, 64\}$. Gran-DAG is trained using the RmsProp optimizer (Tieleman & Hinton, 2012). This model can be used for consequence prediction from learned causal mechanisms. Indeed, it uses an architecture similar to DP-DAG which aims at predicting the values of consequence values from values of the learned parents. We use the official implementation provided in `https://github.com/kurowasan/GraN-DAG`.

**Masked-NN (Ng et al., 2019):** Masked-NN proposes to learn the binary adjacency matrix $A$ by combining the Gumbel-Softmax distribution with the augmented Lagrangian approach for optimization. We follow the recommendation for augmented Lagrangian optimization in (Lachapelle et al., 2020; Ng et al., 2019; Yu et al., 2019; Zheng et al., 2018) and set $\eta = 10$ and $\gamma = \frac{1}{4}$. We used a neural network with 3 linear layers. Further, we performed a grid-search on the learning rate $lr \in [1e^{-4}, 1e^{-2}]$ and the number of hidden units per linear layer $h \in \{8, 16, 64\}$. Masked-NN is trained using the Adam optimizer (Kingma & Ba, 2015). This model can be used for consequence prediction from learned causal mechanisms. Indeed, it uses an architecture similar to DP-DAG which aims at predicting the values of consequence values from values of the learned parents.

**GT-DAG:** GT-DAG is a new baseline with the exact same architecture as VI-DP-DAG except that it uses the ground-truth DAG $A^*$ instead of DAGs sampled from DP-DAG during learning. Hence, this baseline can be considered as the best-case scenario when learning the causal mechanisms $f_{i,\psi}$ from observational data. Further, we performed a grid-search on the learning rate $lr \in [1e^{-4}, 1e^{-2}]$ and the number of hidden units per linear layer $h \in \{8, 16, 64\}$. GT-DAG is trained using the Adam optimizer (Kingma & Ba, 2015). We perform early stopping by checking loss improvement on the validation set every two epochs and a patience $p = 10$.

**CAM (Bühlmann et al., 2014):** CAM proposes to learns DAGs structure by (1) using Preliminary Neighbor Selection (PNS), (2) greedily searching for edges with the largest Likelihodd gain, and finally (3) pruning spurious edges with significance testing of covariates (Bühlmann et al., 2014). The preprocessing PNS step and the postprocessing steps are not differentiable but are decisive for high performance. This motivates other approaches like Lachapelle et al. (2020); Ng et al. (2019) to use the same processing steps.

# F ADDITIONAL RESULTS

## F.1 DAG SAMPLING

In this section, we compare the DAG sampling performances of DP-DAG using Gumbel-Sinkhorn or Gumbel-Top-k parametrization for permutation sampling. To this end, we sample 30 DAGs with 10 to 200 nodes and compute the mean and the variance of the sampling time. We show results for the sampling time in Fig. 4. The Gumbel-Top-k parametrization is significantly faster than the Gumbel-Sinkhorn parametrization. This is expected from the complexity analysis of the two permutation parametrization. Gumbel-Sinkhorn has a complexity of $\mathcal{O}(n^3)$ while Gumbel-Top-k has a complexity of $\mathcal{O}(n^2)$.

Further, we evaluate the capacity of the two differentiable sampling methods for learning DAGs when the ground-truth DAG is observed. To this end, the direct MSE error between a new sampled DAG **A** and the ground-truth DAG $\mathbf{A}^*$ is used as the loss at each iteration. This task aims at evaluating

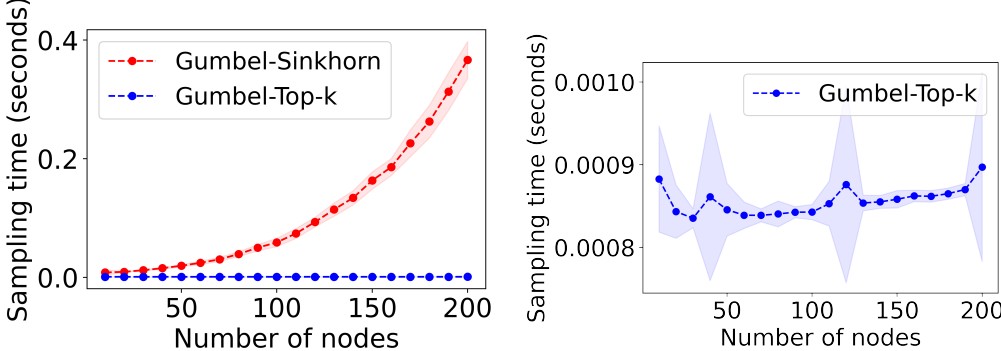

Figure 4: DAG sampling time of DP-DAG using Gumbel-Sinkhorn or Gumbel-Top-k parametrization for permutation sampling. The Gumbel-Top-k parametrization is significantly faster for sampling.

whether the DAG sampling optimization scheme is a reliable optimization scheme on a simple toy task. This is a priori not guaranteed since the DAG sampling space is very large. However, we observe in Tab. 6 that both Gumbel-Sinkhorn and Gumbel-Top-k parametrizations recover almost perfectly the ground-truth DAG $\mathbf{A}^*$ for all graph sizes and graph types. These results are particularly significant since each dataset setting is averaged over 10 sampled graphs and 4 learning rates $lr \in \{10e^{-1}, 10e^{-2}, 10e^{-3}, 10e^{-4}\}$. These experiments validate that DP-DAG is a reliable method for *fast* and *differentiable* DAG sampling.

|  | Gumbel-Sinkhorn | Gumbel-Top-k |
|---|---|---|
| **ER-10-10** | $1.0 \pm 0.0$ | $1.0 \pm 0.0$ |
| **ER-10-40** | $0.97 \pm 0.02$ | $0.98 \pm 0.0$ |
| **ER-20-80** | $0.95 \pm 0.04$ | $0.98 \pm 0.03$ |
| **ER-50-50** | $0.94 \pm 0.06$ | $0.95 \pm 0.05$ |
| **ER-100-100** | $0.9 \pm 0.1$ | $0.96 \pm 0.05$ |
| **ER-100-200** | $0.92 \pm 0.06$ | $0.97 \pm 0.03$ |
| **ER-100-400** | $0.91 \pm 0.06$ | $0.95 \pm 0.05$ |
| **SF-10-10** | $0.87 \pm 0.04$ | $0.92 \pm 0.08$ |
| **SF-10-40** | $0.95 \pm 0.03$ | $0.99 \pm 0.01$ |
| **SF-20-20** | $0.89 \pm 0.06$ | $0.94 \pm 0.09$ |
| **SF-20-80** | $0.98 \pm 0.03$ | $1.0 \pm 0.01$ |
| **SF-50-50** | $0.92 \pm 0.06$ | $0.95 \pm 0.06$ |
| **SF-50-200** | $0.97 \pm 0.03$ | $0.98 \pm 0.02$ |
| **SF-100-100** | $0.89 \pm 0.1$ | $0.94 \pm 0.07$ |
| **SF-100-400** | $0.96 \pm 0.04$ | $0.99 \pm 0.02$ |

Table 4: AUC-PR

|  | Gumbel-Sinkhorn | Gumbel-Top-k |
|---|---|---|
| **ER-10-10** | $1.0 \pm 0.0$ | $1.0 \pm 0.0$ |
| **ER-10-40** | $0.97 \pm 0.0$ | $0.99 \pm 0.0$ |
| **ER-20-80** | $0.96 \pm 0.0$ | $0.98 \pm 0.0$ |
| **ER-50-50** | $0.96 \pm 0.0$ | $0.97 \pm 0.0$ |
| **ER-100-100** | $0.93 \pm 0.01$ | $0.97 \pm 0.0$ |
| **ER-100-200** | $0.94 \pm 0.0$ | $0.98 \pm 0.0$ |
| **ER-100-400** | $0.94 \pm 0.0$ | $0.97 \pm 0.0$ |
| **SF-10-10** | $0.9 \pm 0.0$ | $0.94 \pm 0.01$ |
| **SF-10-40** | $0.96 \pm 0.0$ | $0.99 \pm 0.0$ |
| **SF-20-20** | $0.92 \pm 0.0$ | $0.95 \pm 0.01$ |
| **SF-20-80** | $0.98 \pm 0.0$ | $1.0 \pm 0.0$ |
| **SF-50-50** | $0.95 \pm 0.0$ | $0.97 \pm 0.0$ |
| **SF-50-200** | $0.98 \pm 0.0$ | $0.99 \pm 0.0$ |
| **SF-100-100** | $0.92 \pm 0.01$ | $0.96 \pm 0.0$ |
| **SF-100-400** | $0.97 \pm 0.0$ | $0.99 \pm 0.0$ |

Table 5: AUC-ROC

Table 6: Direct DAG structure learning results on synthetic datasets using differentiable sampling optimization with AUC-PR and AUC-ROC scores (Higher is better). At each iteration, the direct MSE error between a new sampled DAG $\mathbf{A}$ and the ground-truth DAG $\mathbf{A}*$ is used as loss. Each results is averaged over 10 sampled graphs and 4 learning rates. Both Gumbel-Sinkhorn and Gumbel-Top-k sampling methods lead to high performance for all graph sizes and graph types.

## F.2 DAG STRUCTURE LEARNING WITHOUT NON-DIFFERENTIABLE PROCESSING

We show the additional results for *differentiable* DAG structure learning on synthetic and real datasets in Tab. 7. While GraN-DAG competes with VI-DP-DAG on small synthetic datasets with around 10 nodes, DP-DAG outperforms all baselines on all larger synthetic datasets with 50 or 100 nodes for both ER and SF graph types. On real datasets, no baseline consistently outperforms VI-DP-DAG which obtains 4/8 best scores and second-best scores on the remaining evaluations.

Furthermore, we show results for DAG structure learning for VI-DP-DAG, GraN-DAG and Masked-DAG using the mean and the standard deviation of the SHD scores between the ground-truth dag adjacency matrix $\mathbf{A}^*$ and the thresholded predicted adjacency matrix $\mathbb{P}_\phi(U_{\pi(i)\pi(j)}) > t$ or $A_{ij} > t$ for different threshold choices in Fig. 5. Thresholds $t$ are ordered from sparser graphs (i.e. larger thresholds) to denser graphs (i.e. smaller thresholds). We observe that models might be sensitive to

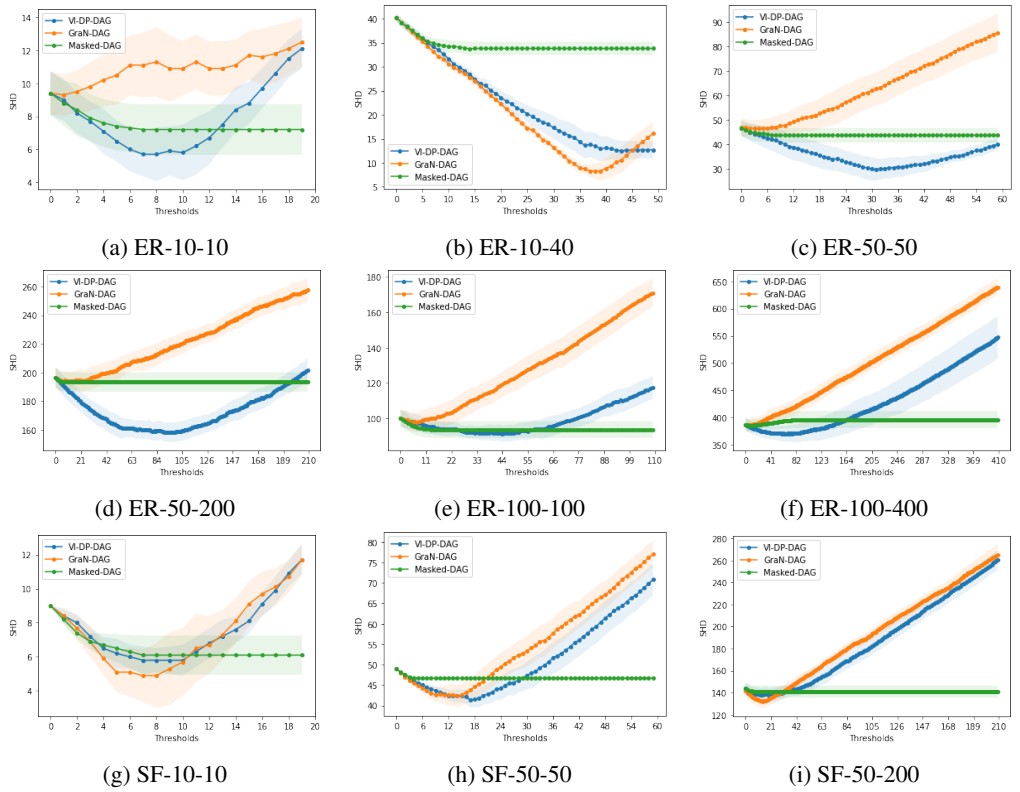

Figure 5: DAG structure learning results for different graph types and sizes for all differentiable VI-DP-DAG, GraN-DAG and Masked-DAG with SHD scores with different thresholds from sparser to denser graphs (Lower is better).

the threshold choice thus motivating the use of threshold agnostic metrics such that AUC-PR and AUC-ROC. However, we observe that VI-DP-DAG generally achieves the best performances for different graphs types and graph sizes given the best possible threshold selection.

### F.3 DAG STRUCTURE LEARNING WITH NON-DIFFERENTIABLE PROCESSING

We show the additional results for DAG structure learning with *non-differentiable* pre- and post-processing in Tab. 8 and Tab. 9. In these experiments, we apply the default pre-processing PNS and post-processing CAM pruning steps to CAM, GraN-DAG, Masked-DAG as suggested by the original papers. We similarly apply CAM pruning to the final results of VI-DP-DAG. While the greedy CAM is the best performing model for most synthetic datasets, VI-DP-DAG maintains competitive performance on all datasets and is the best performing model on real datasets (i.e. 7/8 best scores). The pre- and post-processing steps are decisive for the augmented Lagrangian optimization baselines GraN-DAG and Masked-DAG to achieve performance competitive with VI-DP-DAG. Lachapelle et al. (2020) makes similar observations on the dependence on PNS and CAM pruning steps. We show the comparison of GraN-DAG and VI-DP-DAG with and without additional processing steps in Tab. 10 and Tab. 11. GraN-DAG is indeed more dependent on additional processing steps than VI-DP-DAG to achieve high performance.

### F.4 NON-DIFFERENTIABLE PROCESSING TIME

We show additional results of the processing time (incl. the mean and standard error of the mean over 5 runs) for the non-differentiable PNS pre-processing, the CAM algorithm and the DAG pruning post-processing (Bühlmann et al., 2014) for ER-10-40, ER-50-200, ER-100-400 with with different number of nodes in Tab. 12. Each of this processing step becomes significantly slower tha VI-DP-DAG for larger number of nodes. In particular on ER-100-400, the PNS pre-processing is around ×4

| Models | Un-AUC-PR | Un-AUC-ROC | Dir-AUC-PR | Dir-AUC-ROC |
|---|---|---|---|---|
| **DAG-GNN** | $44.20 \pm 1.66$ | $63.55 \pm 1.06$ | $18.35 \pm 0.9$ | $51.59 \pm 1.0$ |
| **DAG-NF** | $28.22 \pm 0.88$ | $54.07 \pm 0.63$ | $19.47 \pm 0.88$ | $55.41 \pm 0.61$ |
| **GraN-DAG** | $\mathbf{91.17} \pm 0.69$ | $\mathbf{96.76} \pm 0.29$ | $\mathbf{89.96} \pm 0.69$ | $\mathbf{97.38} \pm 0.25$ |
| **Masked-DAG** | $44.74 \pm 1.77$ | $66.58 \pm 1.03$ | $30.23 \pm 1.8$ | $62.28 \pm 1.03$ |
| **VI-DP-DAG** | $\underline{82.88} \pm 1.48$ | $\underline{95.00} \pm 0.51$ | $\underline{62.61} \pm 2.84$ | $\underline{83.27} \pm 1.36$ |

(a) ER-10-10

| Models | Un-AUC-PR | Un-AUC-ROC | Dir-AUC-PR | Dir-AUC-ROC |
|---|---|---|---|---|
| **DAG-GNN** | $85.31 \pm 0.53$ | $55.86 \pm 0.69$ | $49.26 \pm 0.79$ | $55.76 \pm 0.67$ |
| **DAG-NF** | $83.46 \pm 0.29$ | $56.72 \pm 0.43$ | $42.86 \pm 0.23$ | $51.93 \pm 0.43$ |
| **GraN-DAG** | $\mathbf{96.47} \pm 0.14$ | $\mathbf{89.20} \pm 0.38$ | $\mathbf{93.54} \pm 0.39$ | $\mathbf{95.26} \pm 0.34$ |
| **Masked-DAG** | $84.05 \pm 0.3$ | $58.94 \pm 0.31$ | $48.92 \pm 0.42$ | $57.19 \pm 0.35$ |
| **VI-DP-DAG** | $\underline{94.59} \pm 0.5$ | $\underline{86.09} \pm 1.07$ | $\underline{79.42} \pm 0.97$ | $\underline{83.93} \pm 0.76$ |

(b) ER-10-40

| Models | Un-AUC-PR | Un-AUC-ROC | Dir-AUC-PR | Dir-AUC-ROC |
|---|---|---|---|---|
| **DAG-GNN** | $16.20 \pm 1.03$ | $52.20 \pm 0.78$ | $5.52 \pm 0.33$ | $51.80 \pm 0.6$ |
| **DAG-NF** | $4.48 \pm 0.13$ | $50.32 \pm 0.08$ | $2.35 \pm 0.09$ | $50.22 \pm 0.05$ |
| **GraN-DAG** | $\underline{42.97} \pm 1.1$ | $\underline{71.40} \pm 0.7$ | $\underline{38.07} \pm 0.97$ | $\underline{70.27} \pm 0.65$ |
| **Masked-DAG** | $11.29 \pm 0.46$ | $53.97 \pm 0.24$ | $7.66 \pm 0.4$ | $53.26 \pm 0.22$ |
| **VI-DP-DAG** | $\mathbf{75.70} \pm 0.8$ | $\mathbf{97.26} \pm 0.12$ | $\mathbf{52.95} \pm 0.95$ | $\mathbf{80.85} \pm 0.46$ |

(c) ER-50-50

| Models | Un-AUC-PR | Un-AUC-ROC | Dir-AUC-PR | Dir-AUC-ROC |
|---|---|---|---|---|
| **DAG-GNN** | $7.46 \pm 0.24$ | $49.14 \pm 0.39$ | $2.85 \pm 0.16$ | $50.43 \pm 0.59$ |
| **DAG-NF** | $2.12 \pm 0.03$ | $50.13 \pm 0.09$ | $1.10 \pm 0.02$ | $50.12 \pm 0.06$ |
| **GraN-DAG** | $\underline{19.67} \pm 0.58$ | $\underline{64.47} \pm 0.5$ | $\underline{11.63} \pm 0.46$ | $\underline{60.49} \pm 0.38$ |
| **Masked-DAG** | $8.78 \pm 0.31$ | $53.46 \pm 0.16$ | $7.41 \pm 0.32$ | $53.29 \pm 0.16$ |
| **VI-DP-DAG** | $\mathbf{48.38} \pm 0.88$ | $\mathbf{94.46} \pm 0.16$ | $\mathbf{29.34} \pm 0.93$ | $\mathbf{72.93} \pm 0.55$ |

(d) ER-100-100

| Models | Un-AUC-PR | Un-AUC-ROC | Dir-AUC-PR | Dir-AUC-ROC |
|---|---|---|---|---|
| **DAG-GNN** | $38.91 \pm 1.56$ | $46.56 \pm 1.56$ | $20.04 \pm 0.62$ | $49.83 \pm 0.63$ |
| **DAG-NF** | $24.03 \pm 0.87$ | $51.38 \pm 0.81$ | $11.78 \pm 0.46$ | $50.93 \pm 0.39$ |
| **GraN-DAG** | $\mathbf{93.14} \pm 0.75$ | $\mathbf{97.59} \pm 0.28$ | $\mathbf{93.01} \pm 0.77$ | $\mathbf{98.85} \pm 0.14$ |
| **Masked-DAG** | $48.96 \pm 1.79$ | $69.21 \pm 1.05$ | $39.42 \pm 2.22$ | $66.92 \pm 1.2$ |
| **VI-DP-DAG** | $\underline{77.26} \pm 0.97$ | $\underline{93.47} \pm 0.4$ | $\underline{66.35} \pm 1.69$ | $\underline{87.05} \pm 1.18$ |

(e) SF-10-10

| Models | Un-AUC-PR | Un-AUC-ROC | Dir-AUC-PR | Dir-AUC-ROC |
|---|---|---|---|---|
| **DAG-GNN** | $7.76 \pm 0.26$ | $49.24 \pm 0.58$ | $3.36 \pm 0.14$ | $49.66 \pm 0.52$ |
| **DAG-NF** | $3.92 \pm 0.0$ | $49.83 \pm 0.0$ | $1.96 \pm 0.0$ | $49.92 \pm 0.0$ |
| **GraN-DAG** | $\underline{21.82} \pm 0.64$ | $\underline{64.16} \pm 0.51$ | $\underline{18.12} \pm 0.66$ | $\mathbf{62.78} \pm 0.47$ |
| **Masked-DAG** | $10.39 \pm 0.23$ | $53.37 \pm 0.12$ | $6.11 \pm 0.22$ | $52.33 \pm 0.1$ |
| **VI-DP-DAG** | $\mathbf{46.67} \pm 0.8$ | $\mathbf{80.50} \pm 0.49$ | $\mathbf{23.25} \pm 0.89$ | $\underline{60.74} \pm 0.7$ |

(f) SF-50-50

Table 7: DAG structure learning results on synthetic and real datasets for all differentiable DAG learning models with AUC-PR and AUC-ROC scores (Higher is better). *Un-* and *Dir-* indicate scores for undirected and directed edges. Best scores among all models are in bold. Second best scores among all models are underlined.

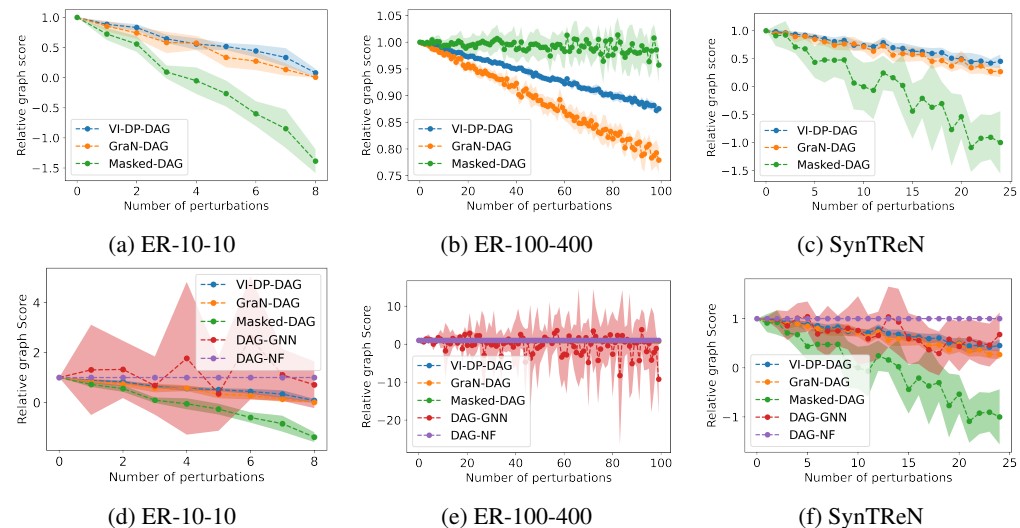

Figure 6: Relative confidence scores of perturbed graphs $\tilde{\mathbf{A}}^*$ obtained by randomly moving edges from the ground-truth DAG adjacency matrix $\mathbf{A}^*$. Fig. 6a-6c show the results for models with architecture similar to VI-DP-DAG. Fig. 6d-6f show the results for models

slower than VI-DP-DAG, the CAM algorithm is at least more than $\times 900$ slower than VI-DP-DAG since it did not finish in less than 2 days, and the DAG pruning post-processing is around $\times 23$ slower than VI-DP-DAG.

## F.5 DAG STRUCTURE LEARNING UNDER PERTURBATIONS.

We evaluate if the models confidently assign lower confidence scores to perturbed versions of the ground-truth adjacency matrix $\mathbf{A}^*$. To this end, we create a perturbed matrix $\tilde{\mathbf{A}}^*$ by randomly moving different number of edges from $\mathbf{A}^*$, and compute its relative confidence score. The relative confidence score assigned by a model is the sum of the edges scores of the perturbed matrix $\tilde{\mathbf{A}}^*$ normalized by the score of the clean matrix $\mathbf{A}^*$ i.e. $\frac{\sum_{ij} S_{ij} \mathbb{1}_{\tilde{A}_{ij}}}{\sum_{ij} S_{ij} \mathbb{1}_{\tilde{A}^*_{ij}}}$ where $S_{ij} = \mathbb{P}_\phi(U_{\pi(i)\pi(j)})$ for VI-DP-DAG and $S_{ij} = A_{ij}$ for other models using Lagrangian optimization.

We report the mean and standard deviation of the relative confidence score over 10 sampled perturbed graphs in Fig. 6. As desired, VI-DP-DAG confidently assigns a higher score to the ground-truth matrix and lower scores to graphs with larger perturbations. This indicates that the DAG sampling optimization of DP-DAG converges to a reliable probabilistic model over DAGs. In contrast, GraN-DAG and Masked-DAG assign significantly more noisy scores to perturbed graphs (see e.g. Fig. 6b). Further, other baselines do not confidently assign strictly decreasing scores to graphs with larger perturbations.

## F.6 CAUSAL MECHANISMS LEARNING

We show causal mechanisms learning results when varying the threshold $t$ used to compute the adjacency matrix $\mathbf{A}$ from the edge probabilities $A_{ij} = \mathbb{1}_{\mathbb{P}_\phi(U_{ij}|\mathbf{\Pi})>t}$ of VI-DP-DAG in Fig. 7. The MSE scores of VI-DP-DAG are almost insensitive to the threshold value except for probability thresholds close to 1. Intuitively, DP-DAG predicts a too sparse DAG adjacency matrix for threshold close to 1, thus removing important Granger-causes to predict the node values $x_i$.

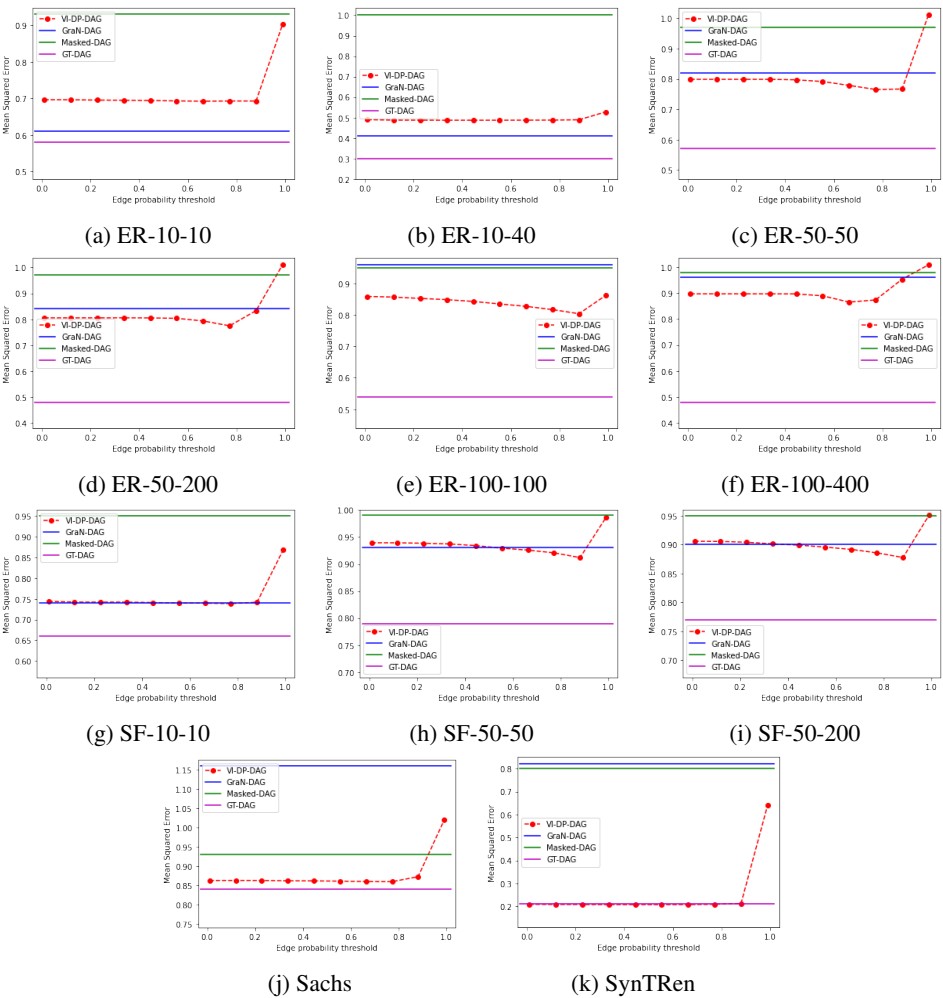

Figure 7: Causal mechanisms results on all datasets with MSE score (Lower is better) on a test set i.e. $||\mathbf{X} - \hat{\mathbf{X}}||_2^2$ where $\hat{X}_i = f_{i,\theta}(\mathbf{A}_i \circ \mathbf{X})$. We vary the threshold $t$ to compute the adjacency matrix $\mathbf{A}$ from the edge probabilities $A_{ij} = \mathbb{1}_{\mathbb{P}_\phi(U_{ij}|\mathbf{\Pi}) > t}$ of VI-DP-DAG. For indication, the MSE of GraN-DAG, Masked-DAG and GT-DAG are plotted with straight lines.

| Models | Un-AUC-PR | Un-AUC-ROC | Dir-AUC-PR | Dir-AUC-ROC |
|---|---|---|---|---|
| **CAM\*** | **92.60** ± 0.17 | **98.89** ± 0.0 | **92.56** ± 0.17 | **98.96** ± 0.0 |
| **GraN-DAG\*** | 87.45 ± 0.76 | 93.51 ± 0.4 | 86.63 ± 0.81 | 93.25 ± 0.41 |
| **Masked-DAG\*** | 11.34 ± 0.45 | 53.97 ± 0.24 | 7.71 ± 0.4 | 53.27 ± 0.22 |
| **VI-DP-DAG\*** | 90.28 ± 0.46 | 95.93 ± 0.24 | 61.13 ± 0.87 | 85.21 ± 0.38 |

(a) ER-50-50

| Models | Un-AUC-PR | Un-AUC-ROC | Dir-AUC-PR | Dir-AUC-ROC |
|---|---|---|---|---|
| **CAM\*** | 49.35 ± 0.05 | **74.45** ± 0.04 | 41.35 ± 0.06 | **73.60** ± 0.04 |
| **GraN-DAG\*** | 49.85 ± 0.66 | 70.91 ± 0.44 | **44.78** ± 0.74 | 70.78 ± 0.44 |
| **Masked-DAG\*** | 17.55 ± 0.14 | 51.09 ± 0.05 | 9.14 ± 0.09 | 50.81 ± 0.04 |
| **VI-DP-DAG\*** | **53.77** ± 0.36 | 73.25 ± 0.25 | 36.54 ± 0.34 | 68.08 ± 0.2 |

(b) ER-50-200

| Models | Un-AUC-PR | Un-AUC-ROC | Dir-AUC-PR | Dir-AUC-ROC |
|---|---|---|---|---|
| **CAM\*** | **56.00** ± 0.0 | **79.46** ± 0.0 | **49.97** ± 0.0 | **78.24** ± 0.0 |
| **GraN-DAG\*** | 38.28 ± 0.39 | 66.95 ± 0.24 | 35.39 ± 0.4 | 66.85 ± 0.24 |
| **Masked-DAG\*** | 9.03 ± 0.12 | 50.72 ± 0.06 | 4.76 ± 0.1 | 50.55 ± 0.06 |
| **VI-DP-DAG\*** | 44.45 ± 0.61 | 72.71 ± 0.23 | 22.32 ± 0.93 | 64.57 ± 0.39 |

(c) ER-100-400

| Models | Un-AUC-PR | Un-AUC-ROC | Dir-AUC-PR | Dir-AUC-ROC |
|---|---|---|---|---|
| **CAM\*** | **53.66** ± 0.16 | **79.43** ± 0.0 | **52.86** ± 0.16 | **79.51** ± 0.0 |
| **GraN-DAG\*** | 49.86 ± 0.99 | 73.96 ± 0.52 | 47.71 ± 0.99 | 73.46 ± 0.5 |
| **Masked-DAG\*** | 10.39 ± 0.23 | 53.37 ± 0.12 | 6.11 ± 0.22 | 52.33 ± 0.1 |
| **VI-DP-DAG\*** | 50.56 ± 0.96 | 74.78 ± 0.52 | 25.41 ± 0.97 | 65.99 ± 0.54 |

(d) SF-50-50

| Models | Un-AUC-PR | Un-AUC-ROC | Dir-AUC-PR | Dir-AUC-ROC |
|---|---|---|---|---|
| **CAM\*** | **32.45** ± 0.13 | **63.10** ± 0.04 | **28.45** ± 0.13 | **63.17** ± 0.04 |
| **GraN-DAG\*** | 29.28 ± 0.41 | 60.11 ± 0.24 | 24.74 ± 0.44 | 60.12 ± 0.24 |
| **Masked-DAG\*** | 14.39 ± 0.13 | 51.64 ± 0.06 | 7.23 ± 0.12 | 50.97 ± 0.06 |
| **VI-DP-DAG\*** | 30.17 ± 0.29 | 60.85 ± 0.17 | 15.78 ± 0.29 | 57.04 ± 0.17 |

(e) SF-50-200

| Models | Un-AUC-PR | Un-AUC-ROC | Dir-AUC-PR | Dir-AUC-ROC |
|---|---|---|---|---|
| **CAM\*** | 32.62 ± 0.04 | 55.07 ± 0.06 | 18.69 ± 0.03 | **56.98** ± 0.02 |
| **GraN-DAG\*** | 30.79 ± 0.2 | 52.07 ± 0.21 | 16.38 ± 0.2 | 52.22 ± 0.26 |
| **Masked-DAG\*** | 36.98 ± 0.25 | 55.13 ± 0.15 | 17.74 ± 0.31 | 51.71 ± 0.23 |
| **VI-DP-DAG\*** | **40.29** ± 0.4 | **59.00** ± 0.06 | **19.11** ± 0.44 | 54.30 ± 0.33 |

(f) Sachs

| Models | Un-AUC-PR | Un-AUC-ROC | Dir-AUC-PR | Dir-AUC-ROC |
|---|---|---|---|---|
| **CAM\*** | 18.27 ± 0.03 | 63.16 ± 0.03 | 7.57 ± 0.01 | 56.16 ± 0.01 |
| **GraN-DAG\*** | 16.30 ± 0.34 | 54.91 ± 0.21 | 9.55 ± 0.24 | 53.77 ± 0.19 |
| **Masked-DAG\*** | 17.24 ± 0.4 | 54.22 ± 0.32 | 8.78 ± 0.42 | 52.63 ± 0.38 |
| **VI-DP-DAG\*** | **35.11** ± 0.91 | **76.23** ± 0.68 | **17.84** ± 0.98 | **64.32** ± 1.05 |

(g) SynTReN

Table 8: DAG structure learning results on synthetic and real datasets for all differentiable DAG learning models with non-differentiable processing denoted with a star *. Performance are evaluated with the AUC-PR and AUC-ROC scores (Higher is better). *Un-* and *Dir-* indicate scores for undirected and directed edges. Best scores among all models are in bold. Second best scores among all models are underlined.

| Models | Un-AUC-PR | Un-AUC-ROC | Dir-AUC-PR | Dir-AUC-ROC |
|---|---|---|---|---|
| **CAM*** | **100.00** $\pm$ 0.0 | **100.00** $\pm$ 0.0 | **100.00** $\pm$ 0.0 | **100.00** $\pm$ 0.0 |
| **GraN-DAG*** | 94.80 $\pm$ 0.76 | 98.68 $\pm$ 0.22 | 86.34 $\pm$ 1.92 | 95.96 $\pm$ 0.55 |
| **Masked-DAG*** | 45.56 $\pm$ 1.64 | 66.35 $\pm$ 1.02 | 29.69 $\pm$ 1.78 | 61.96 $\pm$ 1.0 |
| **VI-DP-DAG*** | 88.17 $\pm$ 1.07 | 96.18 $\pm$ 0.43 | 66.17 $\pm$ 2.66 | 87.58 $\pm$ 1.0 |

(a) ER-10-10

| Models | Un-AUC-PR | Un-AUC-ROC | Dir-AUC-PR | Dir-AUC-ROC |
|---|---|---|---|---|
| **CAM*** | 89.56 $\pm$ 0.0 | 78.09 $\pm$ 0.0 | 79.56 $\pm$ 0.0 | 85.54 $\pm$ 0.0 |
| **GraN-DAG*** | **93.79** $\pm$ 0.38 | **84.43** $\pm$ 0.78 | **86.00** $\pm$ 0.55 | **89.28** $\pm$ 0.37 |
| **Masked-DAG*** | 84.00 $\pm$ 0.31 | 58.82 $\pm$ 0.32 | 48.77 $\pm$ 0.43 | 57.06 $\pm$ 0.36 |
| **VI-DP-DAG*** | 93.23 $\pm$ 0.46 | 83.11 $\pm$ 0.92 | 75.98 $\pm$ 0.83 | 80.75 $\pm$ 0.62 |

(b) ER-10-40

| Models | Un-AUC-PR | Un-AUC-ROC | Dir-AUC-PR | Dir-AUC-ROC |
|---|---|---|---|---|
| **CAM*** | **82.17** $\pm$ 0.0 | **95.40** $\pm$ 0.0 | 75.07 $\pm$ 0.0 | **93.43** $\pm$ 0.0 |
| **GraN-DAG*** | 77.87 $\pm$ 0.47 | 90.08 $\pm$ 0.28 | **76.61** $\pm$ 0.53 | 89.71 $\pm$ 0.29 |
| **Masked-DAG*** | 8.78 $\pm$ 0.31 | 53.46 $\pm$ 0.16 | 7.41 $\pm$ 0.32 | 53.29 $\pm$ 0.16 |
| **VI-DP-DAG*** | 79.39 $\pm$ 0.57 | 93.78 $\pm$ 0.21 | 46.11 $\pm$ 1.13 | 80.34 $\pm$ 0.37 |

(c) ER-100-100

| Models | Un-AUC-PR | Un-AUC-ROC | Dir-AUC-PR | Dir-AUC-ROC |
|---|---|---|---|---|
| **CAM*** | **100.00** $\pm$ 0.0 | **100.00** $\pm$ 0.0 | **100.00** $\pm$ 0.0 | **100.00** $\pm$ 0.0 |
| **GraN-DAG*** | 90.89 $\pm$ 1.15 | 94.44 $\pm$ 0.7 | 86.85 $\pm$ 1.36 | 93.79 $\pm$ 0.68 |
| **Masked-DAG*** | 48.07 $\pm$ 1.63 | 68.25 $\pm$ 1.0 | 38.96 $\pm$ 1.99 | 66.48 $\pm$ 1.1 |
| **VI-DP-DAG*** | 83.76 $\pm$ 0.89 | 90.65 $\pm$ 0.52 | 71.25 $\pm$ 1.65 | 85.52 $\pm$ 0.81 |

(d) SF-10-10

Table 9: DAG structure learning results on synthetic and real datasets for all differentiable DAG learning models with non-differentiable processing denoted with a star *. Performance are evaluated with the AUC-PR and AUC-ROC scores (Higher is better). *Un-* and *Dir-* indicate scores for undirected and directed edges. Best scores among all models are in bold. Second best scores among all models are underlined.

| Models | Un-AUC-PR | Un-AUC-ROC | Dir-AUC-PR | Dir-AUC-ROC |
|---|---|---|---|---|
| **GraN-DAG** | $42.97 \pm 1.1$ | $71.40 \pm 0.7$ | $38.07 \pm 0.97$ | $70.27 \pm 0.65$ |
| **GraN-DAG\*** | $87.45 \pm 0.76$ | $93.51 \pm 0.4$ | $86.63 \pm 0.81$ | $93.25 \pm 0.41$ |
| **VI-DP-DAG** | $75.70 \pm 0.8$ | $97.26 \pm 0.12$ | $52.95 \pm 0.95$ | $80.85 \pm 0.46$ |
| **VI-DP-DAG\*** | $90.28 \pm 0.46$ | $95.93 \pm 0.24$ | $61.13 \pm 0.87$ | $85.21 \pm 0.38$ |

(a) ER-50-50

| Models | Un-AUC-PR | Un-AUC-ROC | Dir-AUC-PR | Dir-AUC-ROC |
|---|---|---|---|---|
| **GraN-DAG** | $25.11 \pm 0.4$ | $56.57 \pm 0.35$ | $17.61 \pm 0.47$ | $56.35 \pm 0.34$ |
| **GraN-DAG\*** | $49.85 \pm 0.66$ | $70.91 \pm 0.44$ | $44.78 \pm 0.74$ | $70.78 \pm 0.44$ |
| **VI-DP-DAG** | $62.08 \pm 0.34$ | $84.53 \pm 0.25$ | $41.70 \pm 0.39$ | $71.08 \pm 0.33$ |
| **VI-DP-DAG\*** | $53.77 \pm 0.36$ | $73.25 \pm 0.25$ | $36.54 \pm 0.34$ | $68.08 \pm 0.2$ |

(b) ER-50-200

| Models | Un-AUC-PR | Un-AUC-ROC | Dir-AUC-PR | Dir-AUC-ROC |
|---|---|---|---|---|
| **GraN-DAG** | $9.73 \pm 0.11$ | $52.82 \pm 0.11$ | $4.42 \pm 0.04$ | $51.04 \pm 0.06$ |
| **GraN-DAG\*** | $38.28 \pm 0.39$ | $66.95 \pm 0.24$ | $35.39 \pm 0.4$ | $66.85 \pm 0.24$ |
| **VI-DP-DAG** | $36.98 \pm 0.98$ | $77.23 \pm 0.63$ | $19.00 \pm 1.07$ | $60.35 \pm 1.01$ |
| **VI-DP-DAG\*** | $44.45 \pm 0.61$ | $72.71 \pm 0.23$ | $22.32 \pm 0.93$ | $64.57 \pm 0.39$ |

(c) ER-100-400

| Models | Un-AUC-PR | Un-AUC-ROC | Dir-AUC-PR | Dir-AUC-ROC |
|---|---|---|---|---|
| **GraN-DAG** | $21.82 \pm 0.64$ | $64.16 \pm 0.51$ | $18.12 \pm 0.66$ | $62.78 \pm 0.47$ |
| **GraN-DAG\*** | $49.86 \pm 0.99$ | $73.96 \pm 0.52$ | $47.71 \pm 0.99$ | $73.46 \pm 0.5$ |
| **VI-DP-DAG** | $46.67 \pm 0.8$ | $80.50 \pm 0.49$ | $23.25 \pm 0.89$ | $60.74 \pm 0.7$ |
| **VI-DP-DAG\*** | $50.56 \pm 0.96$ | $74.78 \pm 0.52$ | $25.41 \pm 0.97$ | $65.99 \pm 0.54$ |

(d) SF-50-50

| Models | Un-AUC-PR | Un-AUC-ROC | Dir-AUC-PR | Dir-AUC-ROC |
|---|---|---|---|---|
| **GraN-DAG** | $18.75 \pm 0.29$ | $54.76 \pm 0.17$ | $12.35 \pm 0.32$ | $53.31 \pm 0.43$ |
| **GraN-DAG\*** | $29.28 \pm 0.41$ | $60.11 \pm 0.24$ | $24.74 \pm 0.44$ | $60.12 \pm 0.24$ |
| **VI-DP-DAG** | $33.81 \pm 0.39$ | $68.18 \pm 0.39$ | $17.46 \pm 0.34$ | $59.70 \pm 0.65$ |
| **VI-DP-DAG\*** | $30.17 \pm 0.29$ | $60.85 \pm 0.17$ | $15.78 \pm 0.29$ | $57.04 \pm 0.17$ |

(e) SF-50-200

| Models | Un-AUC-PR | Un-AUC-ROC | Dir-AUC-PR | Dir-AUC-ROC |
|---|---|---|---|---|
| **GraN-DAG** | $34.28 \pm 0.32$ | $54.57 \pm 0.36$ | $19.95 \pm 0.35$ | $55.91 \pm 0.33$ |
| **GraN-DAG\*** | $30.79 \pm 0.2$ | $52.07 \pm 0.21$ | $16.38 \pm 0.2$ | $52.22 \pm 0.26$ |
| **VI-DP-DAG** | $43.37 \pm 0.55$ | $59.78 \pm 0.62$ | $22.96 \pm 0.81$ | $60.02 \pm 0.91$ |
| **VI-DP-DAG\*** | $40.29 \pm 0.4$ | $59.00 \pm 0.06$ | $19.11 \pm 0.44$ | $54.30 \pm 0.33$ |

(f) Sachs

| Models | Un-AUC-PR | Un-AUC-ROC | Dir-AUC-PR | Dir-AUC-ROC |
|---|---|---|---|---|
| **GraN-DAG** | $19.89 \pm 0.4$ | $62.29 \pm 0.31$ | $15.39 \pm 0.44$ | $63.91 \pm 0.34$ |
| **GraN-DAG\*** | $16.30 \pm 0.34$ | $54.91 \pm 0.21$ | $9.55 \pm 0.24$ | $53.77 \pm 0.19$ |
| **VI-DP-DAG** | $26.33 \pm 0.56$ | $70.84 \pm 0.45$ | $14.17 \pm 0.76$ | $59.84 \pm 1.46$ |
| **VI-DP-DAG\*** | $35.11 \pm 0.91$ | $76.23 \pm 0.68$ | $17.84 \pm 0.98$ | $64.32 \pm 1.05$ |

(g) SynTReN

Table 10: DAG structure learning results on synthetic and real datasets for VI-DP-DAG and the second best baseline GraN-DAG with and without non-differentiable processing. Models using additional processing are denoted with a star *. Performance are evaluated with the AUC-PR and AUC-ROC scores (Higher is better). *Un-* and *Dir-* indicate scores for undirected and directed edges. GraN-DAG is more dependent on additional processing steps than VI-DP-DAG to achieve high performance.

| Models | Un-AUC-PR | Un-AUC-ROC | Dir-AUC-PR | Dir-AUC-ROC |
|---|---|---|---|---|
| **GraN-DAG** | $91.17 \pm 0.69$ | $96.76 \pm 0.29$ | $89.96 \pm 0.69$ | $97.38 \pm 0.25$ |
| **GraN-DAG*** | $94.80 \pm 0.76$ | $98.68 \pm 0.22$ | $86.34 \pm 1.92$ | $95.96 \pm 0.55$ |
| **VI-DP-DAG** | $82.88 \pm 1.48$ | $95.00 \pm 0.51$ | $62.61 \pm 2.84$ | $83.27 \pm 1.36$ |
| **VI-DP-DAG*** | $88.17 \pm 1.07$ | $96.18 \pm 0.43$ | $66.17 \pm 2.66$ | $87.58 \pm 1.0$ |

(a) ER-10-10

| Models | Un-AUC-PR | Un-AUC-ROC | Dir-AUC-PR | Dir-AUC-ROC |
|---|---|---|---|---|
| **GraN-DAG** | $96.47 \pm 0.14$ | $89.20 \pm 0.38$ | $93.54 \pm 0.39$ | $95.26 \pm 0.34$ |
| **GraN-DAG*** | $93.79 \pm 0.38$ | $84.43 \pm 0.78$ | $86.00 \pm 0.55$ | $89.28 \pm 0.37$ |
| **VI-DP-DAG** | $94.59 \pm 0.5$ | $86.09 \pm 1.07$ | $79.42 \pm 0.97$ | $83.93 \pm 0.76$ |
| **VI-DP-DAG*** | $93.23 \pm 0.46$ | $83.11 \pm 0.92$ | $75.98 \pm 0.83$ | $80.75 \pm 0.62$ |

(b) ER-10-40

| Models | Un-AUC-PR | Un-AUC-ROC | Dir-AUC-PR | Dir-AUC-ROC |
|---|---|---|---|---|
| **GraN-DAG** | $19.67 \pm 0.58$ | $64.47 \pm 0.5$ | $11.63 \pm 0.46$ | $60.49 \pm 0.38$ |
| **GraN-DAG*** | $77.87 \pm 0.47$ | $90.08 \pm 0.28$ | $76.61 \pm 0.53$ | $89.71 \pm 0.29$ |
| **VI-DP-DAG** | $48.38 \pm 0.88$ | $94.46 \pm 0.16$ | $29.34 \pm 0.93$ | $72.93 \pm 0.55$ |
| **VI-DP-DAG*** | $79.39 \pm 0.57$ | $93.78 \pm 0.21$ | $46.11 \pm 1.13$ | $80.34 \pm 0.37$ |

(c) ER-100-100

| Models | Un-AUC-PR | Un-AUC-ROC | Dir-AUC-PR | Dir-AUC-ROC |
|---|---|---|---|---|
| **GraN-DAG** | $93.14 \pm 0.75$ | $97.59 \pm 0.28$ | $93.01 \pm 0.77$ | $98.85 \pm 0.14$ |
| **GraN-DAG*** | $90.89 \pm 1.15$ | $94.44 \pm 0.7$ | $86.85 \pm 1.36$ | $93.79 \pm 0.68$ |
| **VI-DP-DAG** | $77.26 \pm 0.97$ | $93.47 \pm 0.4$ | $66.35 \pm 1.69$ | $87.05 \pm 1.18$ |
| **VI-DP-DAG*** | $83.76 \pm 0.89$ | $90.65 \pm 0.52$ | $71.25 \pm 1.65$ | $85.52 \pm 0.81$ |

(d) SF-10-10

Table 11: DAG structure learning results on synthetic and real datasets for VI-DP-DAG and the second best baseline GraN-DAG with and without non-differentiable processing. Models using additional processing are denoted with a star *. Performance are evaluated with the AUC-PR and AUC-ROC scores (Higher is better). *Un-* and *Dir-* indicate scores for undirected and directed edges. GraN-DAG is more dependent on additional processing steps than VI-DP-DAG to achieve high performance.

| | PNS | CAM | Pruning |
|---|---|---|---|
| **ER-10-40** | $11.91 \pm 0.09$ | $14.42 \pm 0.42$ | $4.29 \pm 0.1$ |
| **ER-50-200** | $150.54 \pm 0.03$ | $22292.7 \pm 591.13$ | $667.03 \pm 15.59$ |
| **ER-100-400** | $750.14 \pm 8.53$ | $> 2$ days | $6114.62 \pm 495.93$ |

Table 12: Processing time in seconds of the non-differentaible PNS pre-processing, the CAM algorthm and the DAG pruning (Bühlmann et al., 2014).

