# OpenReview forum: "Differentiable DAG Sampling"
_ICLR.cc/2022/Conference — ICLR 2022 Poster_

### Official Review · Reviewer_oBzg · 2021-10-30

**Correctness:** 3
**Technical Novelty And Significance:** 2
**Empirical Novelty And Significance:** Not applicable
**Recommendation:** 5
**Confidence:** 4

**Main Review:**

Following the work of NOTEARS, this paper focuses on the problem of DAG structure learning using gradient-based methods. Existing algorithms, such as NOTEARS and Gran-DAG, require minimizing a differentiable loss function with respect to the graph structure $G$ with an equality constraint $h(G)=0$. This equality constraint holds if and only if $G$ is a DAG. However, to solve this problem, one needs to use the augmented Lagrangian method, which usually requires alternatively solving an inner and an outer problem. As mentioned by the authors, “the augmented Lagrangian optimization is computationally expensive”.

The main contribution of this work is to propose a differentiable DAG sampling strategy that can be easily implemented with PyTorch or TensorFlow, and this differentiable DAG sampling strategy can ensure that the searching space when solving the optimization problem is exactly the space of DAGs. Thus, the proposed method does not use the augmented Lagrangian method and is more efficient than most of the current methods.

However, my major concern is that the proposed method of this paper is only a combination of well-developed techniques. The start point of this paper is Theorem 1, which states that any adjacency matrix of a DAG can be obtained by simultaneously applying row and column-wise permutation to an upper triangular matrix. This result is trivial and has been known for a very long time. Thus, I think it is not appropriate to state it as a theorem. Moreover, Theorem 1 has been applied to sampling DAGs already. In fact, many packages use this result to generate random DAGs. Secondly, to make the sampling procedure differentiable, this paper directly uses the recently developed Gumbel-Softmax, Gumbel-Sinkhorn, and Gumbel-Top-k tricks. Therefore, in my opinion, the idea of separately sampling an upper triangular matrix and a permutation matrix in order to sample a DAG is not novel.

Another problem is about Equation (3). It would be better to show that the definition of $P({\bf A})$ is indeed a probability measure.

Finally, in the experiments, the authors used AUC to measure the performance of different methods. This measure might be better than others, but for the completeness of the paper, other commonly used measures should also be included. For example, the ROC curves, SHDs (or SHDs vs. thresholds), and SIDs (or SIDs vs. thresholds).

A minor point: the fourth line of the fourth paragraph on page 3, Thm. 1 -> Th. 1


**Summary Of The Paper:**

Using the recently developed differentiable sampling techniques for discrete variables (e.g. Gumbel-Softmax) and variable orderings (e.g. Gumbel-Sinkhorn and Gumbel-Top-k), this paper proposes a differentiable DAG sampling strategy, and applies it to the problem of DAG learning.

**Summary Of The Review:**

Overall, this paper is well-written. However, the proposed method is only a simple combination of well-developed techniques, and the experiments need to include more metrics. Thus, I think the contributions are only marginally significant or novel.

---

> ### Author Response · Authors · 2021-11-17
> **Author Response to Reviewer oBzg**
>
> We would like to thank you for your valuable comments and suggestions. Furthermore, we would like to provide comments on the novel aspects of our work, the probability measure property of our model and additional structural metrics. Further, we are happy to provide additional clarifications in case you have follow-up questions.
>
> > However, my major concern is that the proposed method of this paper is only a combination of well-developed techniques. The start point of this paper is Theorem 1, which states that any adjacency matrix of a DAG can be obtained by simultaneously applying row and column-wise permutation to an upper triangular matrix. This result is trivial and has been known for a very long time. Thus, I think it is not appropriate to state it as a theorem. Moreover, Theorem 1 has been applied to sampling DAGs already. In fact, many packages use this result to generate random DAGs. Secondly, to make the sampling procedure differentiable, this paper directly uses the recently developed Gumbel-Softmax, Gumbel-Sinkhorn, and Gumbel-Top-k tricks. Therefore, in my opinion, the idea of separately sampling an upper triangular matrix and a permutation matrix in order to sample a DAG is not novel.
>
> **Novelty.** The main novelty of DP-DAG is its capacity to perform *fast and differentiable DAG sampling*. None of the Gumbel-Softmax, Gumbel-Sinkhorn or Gumbel-Top-k tricks alone can differentiably sample DAGs. Thus, the combination of Gumbel-Softmax for edge sampling, Gumbel-Sinkhorn or Gumbel-Top-k/SoftSort for permutation sampling and Thm. 1 is new. We state Thm.1 as a theorem since the matrix formulation is a key component of DP-DAG, thus requiring to be highlighted. We are happy to state it as a "Property", "Lemma" and cite any previous work/packages using the same formulation. Further, the combination of variational inference with the new DP-DAG also leads to the new VI-DP-DAG model which achieves very competitive scores for DAG structure learning and causal mechanisms learning. Therefore, first, the simplicity of (VI-)DP-DAG is a significant technical advantage of our method. Second, the high speed and performance of (VI-)DP-DAG is a significant empirical advantage of our method compared to existing methods.
>
> > Another problem is about Equation (3). It would be better to show that the definition of is indeed a probability measure.
>
> **Probability measure validity.** The fact that the new DP-DAG defined by equation (3) is a valid probability measure follows from the law of total probability i.e.  $P(A)=\sum_{\Pi, U} P(A | U, \Pi) P(U, \Pi) = \sum_{\Pi, U} \mathbb{1}_{A=\Pi^TU\Pi} P(U, \Pi)$ where $U \in \mathcal{U}_n$ and $\Pi \in \mathcal{P}(G)$ are the edge and permutation matrices. Then, assuming that $U$ and $\Pi$ are independent leads to equation (3) which is then a valid probability measure.
>
> > Finally, in the experiments, the authors used AUC to measure the performance of different methods. This measure might be better than others, but for the completeness of the paper, other commonly used measures should also be included. For example, the ROC curves, SHDs (or SHDs vs. thresholds), and SIDs (or SIDs vs. thresholds).
>
> **Experiment metrics.** Our extensive experiment set-up evaluate DAG structure learning, causal mechanisms learning and training speed. In contrast, Lachapelle et al., 2020; Ng et al., 2019; Yuet al., 2019; Zheng et al., 2018 mostly evaluate DAG structure learning. For completeneness, beyond the AUC-PR and AUC-ROC scores, we additionally provide in Fig.4 in the appendix F.2 new results for DAG structure learning using SHD scores with different graph types and sizes and with different threshold choices. Thresholds $t$ are ordered from sparser graphs to denser graphs. We observe that models might be sensitive to the threshold choice thus motivating the use of threshold agnostic metrics such as AUC-PR and AUC-ROC. However, we observe that VI-DP-DAG generally achieves the best performances for different graphs types and graph sizes given the best possible threshold selection.

---

> > ### Comment · Reviewer_oBzg · 2021-11-29
> > **Thank you for the reply**
> >
> > I would like to thank the authors for their responses. About my first concern, the function named randDAG implemented in R package pcalg uses the same methodology to sample DAGs (see, e.g., line 65, https://github.com/cran/pcalg/blob/master/R/genRandDAG.R). Another example is the function named simulate_dag implemented in the NOTEARS package (see, e.g., line 17, https://github.com/xunzheng/notears/blob/master/notears/utils.py). Although the combination of differentiable sampling techniques and Thm. 1 is new, I still think the contribution of the work (which is a sampling procedure for DAGs) is limited. Thus, I will keep my score unchanged.

---

> > > ### Author Response · Authors · 2021-11-29
> > > **Author Response to Reviewer oBzg (2)**
> > >
> > > Thanks a lot for your comment.
> > >
> > > We would like to clarify the contributions of the work which is not limited to a sampling procedure for DAGs (see Sec. 1 paragraph 'contribution'). The first contribution is a **fast** and **differentiable** DAG sampling procedure (DP-DAG) which can be used during optimization. The randDAG in pcalg R package and the simulate_dag in the NOTEARS github repo are designed to generate random synthetic dataset of dags and cannot be used during gradient-based optimization. Further, none of the previous DAG sampling works have similar properties.
> > >
> > > The second contribution is a *new* method for DAG learning from observational data (VI-DP-DAG) which uses DP-DAG and variational inference. VI-DP-DAG guarantees **valid** DAG outputs at **any** time during training. None of the previous differentiable DAG sampling methods have similar properties.
> > >
> > > The third contribution is our extensive experiments which shows that VI-DP-DAG outperforms other differentiable baselines for **DAG structure** and **causal mechanisms** learning and train one order of magnitude **faster**. In contrast,  Lachapelle et al., 2020; Ng et al., 2019; Yuet al., 2019; Zheng et al.,  2018 mostly evaluate on DAG structure learning only.
> > >
> > > Therefore, these three contributions provide significant **technical** and **empirical** advantages compared to existing works.

---

### Official Review · Reviewer_cRyc · 2021-11-02

**Correctness:** 4
**Technical Novelty And Significance:** 3
**Empirical Novelty And Significance:** 3
**Recommendation:** 8
**Confidence:** 3

**Main Review:**

_Strengths_
* The method is simple and clever. Rather than relying on relaxations during the forward pass, the Gumbel Straight-Through estimator allows the relaxation to happen only during gradient estimation. Additionally, the errors in the straight-through estimators for edge and order parameters do not compound due to the probabilistic model.
* The experimental evaluation is thorough. The model is evaluated on both synthetic (Erdos-Renyi, Scale-Free, and SynTReN) and real data (Sachs) of various sizes. The results are convincing, with the proposed method VI-DP-DAG either achieving the best performance or comparable for structure learning. A similar trend holds for causal mechanism learning, where VI-DP-DAG outperforms or ties baselines in all but the two smallest settings.
* Timing results are very impressive. VI-DP-DAG obtains 5x-10x speedup over baselines. In conjunction with the accuracy results, this is a strong contribution.
* The paper is clearly written.

_Weaknesses_
* I did not find any obvious weaknesses.

_Questions_
* How many epochs does each method take to converge? In other words, what is the total training time for each method?
* What are the accuracies and speeds for non-differentiable methods?

_Feedback_
* Can you add a table with DP-DAG and second best method with and without pre/post processing to the appendix? This would make it easier to see the claim that DP-DAG is less dependent on pre/post processing compared to other methods.
* Table 2 (bottom left): GraN-DAG is bolded when it should not be.
* Given the citation of Vowels et. al., 2021 after the discussion of AUC vs Structural Hamming Distance evaluation metrics in the results section, I could not find discussion in Vowels et. al. on a comparison between AUC and SHD. How is this citation relevant in this setting, or did I miss something?


**Summary Of The Paper:**

This paper presents a differentiable probabilistic model (DP-DAG) over DAGs that retains accuracy over comparable differentiable DAG models. In combination with variational inference for training, VI-DP-DAG demonstrates large gains in computational efficiency at the cost of an intractable exact scoring function (ie computing $P_{\phi,\psi}(A)$). The computational gains come from the factorization of the DAG distribution into the product of orderings and edges, which results in intractable scoring as all valid permutations must be marginalized over in order to score a graph. However, this is not an issue: the pathwise derivative is used during training, requiring only a differentiable sampling procedure, and evaluation does not require scoring.

**Summary Of The Review:**

I recommend this paper for acceptance. It proposes a simple method that obtains accuracy competitive to baselines (often exceeding them) at much faster training speeds (between 5x and 10x).

---

> ### Author Response · Authors · 2021-11-17
> **Author Response to Reviewer cRyc**
>
> We would like to thank you for your valuable comments and suggestions. In particular, we appreciate the reviewer's positive feedback and we are eager to provide answers to the reviewer's questions about the total training time evaluation, the performance of the non-differentiable methods and the comparison with/without non-differentiable processings. We are happy to provide additional explanations in case you have follow-up questions.
>
> > How many epochs does each method take to converge? In other words, what is the total training time for each method?
>
> **Total training time.** For a fair comparison, we report the *total training time* in seconds for all methods in Tab.5. VI-DP-DAG is $\times 5$ to $\times 18$ faster than all other differentiable methods. Additionally, we added new results showing the processing time of the PNS pre-processing, the CAM method and the CAM-pruning post-processing in Tab. 15 in the appendix (see 'Performance of non-differentiable methods' in the rebuttal).
>
> > What are the accuracies and speeds for non-differentiable methods?
>
> **Performance of non-differentiable methods.** We provide numbers for DAG structure predictions of non-differentiable methods in Tab.8, 9 in the appendix. The results include GraN-DAG* and Masked-DAG* which uses non-differentiable pre- and post-processing and the CAM* algorithm which is the best performing non-differentiable method accroding to (Lachapelle et al., 2020; Ng et al., 2019). Further, we added new results showing the processing time of the PNS pre-processing, the CAM method and the CAM-pruning post-processing in Tab. 15 in the appendix. Each of this processing step becomes significantly slower than the optimization of VI-DP-DAG for larger number of nodes. In particular on ER-100-400, the PNS pre-processing is around $\times 4$ slower than VI-DP-DAG, the CAM algorithm is at least more than $\times 900$ slower than VI-DP-DAG since it did not finish in less than $2$ days and the DAG pruning post-processing is around $\times 23$ slower than VI-DP-DAG.
>
> > Can you add a table with DP-DAG and second best method with and without pre/post processing to the appendix? This would make it easier to see the claim that DP-DAG is less dependent on pre/post processing compared to other methods.
>
> **Comparison with/without processing.** We added new additional Tab. 13 and Tab. 14 in the appendix comparing VI-DP-DAG and the second best baseline GraN-DAG with and without non-differentiable processing. Indeed, GraN-DAG shows to be more dependent on additional processing steps than VI-DP-DAG to achieve high performance (see e.g. Tab.10 \(c\)). Further, non-differentiable processings significantly slow down the differentiable methods (see Tab. 3 and the new Tab. 15 in the appendix).
>
> > Given the citation of Vowels et. al., 2021 after the discussion of AUC vs Structural Hamming Distance evaluation metrics in the results section, I could not find discussion in Vowels et. al. on a comparison between AUC and SHD. How is this citation relevant in this setting, or did I miss something?
>
> **Clarification AUC vs SHD**. Vowels et al. only present these metrics in section 3.5 'SHD metrics' and 'AUC metrics'. It explains that AUC scores do not depend on a threshold $t$. For completeneness, beyond the AUC-PR and AUC-ROC scores, we additionally provide in Fig.4 in the appendix new results for DAG structure learning using SHD scores with different graph types and sizes and with different threshold choices. Thresholds $t$ are ordered from sparser graphs to denser graphs. We observe that models might be sensitive to the threshold choice thus motivating the use of threshold agnostic metrics such as AUC-PR and AUC-ROC. However, we observe that VI-DP-DAG generally achieves the best performances for different graphs types and graph sizes given the best possible threshold selection.

---

### Official Review · Reviewer_PQoz · 2021-11-03

**Correctness:** 3
**Technical Novelty And Significance:** 3
**Empirical Novelty And Significance:** 2
**Recommendation:** 5
**Confidence:** 4

**Main Review:**

Strengths:
- Contrary to previous work, the proposed method does not pre- or post-processing to make sure the returned graph is DAG.
- It is easy to implement the algorithm in a few lines of code.
- The current work presents a differentiable DAG sampling which I think is novel in this field.

Weaknesses:
- From the experimental results, it is hard to conclude that the current work is better than previous works in terms of both processing time and estimation error. It would be great if the authors can derive the computational complexities of other previous works and compare them with their method. From table 9, we can see that the performances of all methods get better as we use non-differentiable processing and even other methods (like CAM) might achieve higher performance.
- It is unclear whether the DAG sampler is unbiased, i.e., it selects a DAG uniformly from the Markov equivalence class as number of samples goes to infinity.



**Summary Of The Paper:**

The authors proposed an algorithm for sampling DAGs that is suited for continuous optimization. The sampling algorithm has two main steps: In the first step, a causal order over the variables is selected. In the second step, edges are sampled based on the selected order. Moreover, based on this algorithm, they proposed a method in order to learn the causal structure from the observational data. The causal structure learning algorithm is guaranteed to output a DAG at any time and it is not required any pre- or post-processing unlike previous work.

**Summary Of The Review:**

As far as I know, this is the first work, proposing a differentiable DAG sampling. However, it is not clear whether it has some advantages as some post-processing is still needed if one wants to get better performance. Regarding the time complexity, a thorough analysis of other related work is needed. Moreover, it is suggested to compare to some recent works like the following in their experiments:
"DAGs with No Curl: An Efficient DAG Structure Learning Approach", https://arxiv.org/abs/2106.07197

---

> ### Author Response · Authors · 2021-11-17
> **Author Response to Reviewer PQoz**
>
> We would like to thank you for your valuable comments and suggestions. Following your concerns, we would like to provide comments on non-differentiable methods, processing time and performance and the bias property of DAG sampler. We are happy to provide additional clarifications in case you have follow-up questions.
>
> > From the experimental results, it is hard to conclude that the current work is better than previous works in terms of both processing time and estimation error. It would be great if the authors can derive the computational complexities of other previous works and compare them with their method.
>
> **Processing time & Performance.** The processing time to predict the DAG adjacency matrix $A$ from a new dataset $\mathcal{D}$ requires to complete train a new model (see Sec. 5.2 'Training time' p.9). We show that this time is $\times 5$ to $\times 18$ faster for VI-DP-DAG than all other differentiable baselines (see Tab. 3). The computation complexity cannot be derived in closed-form for differentiable methods since it depends on the convergence speed of the gradient-based optimization. The complexity of CAM is $O(p^2)$ where $p$ is the number of possible parents. We show new results in Tab. 12 in the appendix showing that CAM is significantly slower than VI-DP-DAG. In particular on ER-100-400, the CAM algorithm is at least more than $\times 900$ slower than VI-DP-DAG since it did not finish in less than $2$ days. The processing time for consequence predictions corresponds to the time for a single forward pass which has a similar computational complexity for all GraN-DAG, Masked-DAG, GT-DAG and VI-DP-DAG models. Indeed, all these models use the same MLPs architectures $f_{i, \theta}$ to model the causal mechanisms and then predict the value of a node from its parents i.e. $\hat{x_i} = f_{i, \theta}(x_\text{pa}(i))$ (see description 'Causal mechanisms' p.7).
>
> Further, VI-DP-DAG is the best performing differentiable model for estimation error in DAG structure prediction (27/42 best score and second best score otherwise) and consequence predictions (9/11 best score and second best score otherwise) with particularly strong improvement for larger number datasets with $50$ nodes or more (see Tab.1, 2, 7 and Sec.5.2 'DAG structure' & 'Causal Mechanisms' p6-7).
>
> > From table 9, we can see that the performances of all methods get better as we use non-differentiable processing and even other methods (like CAM) might achieve higher performance.
>
> **Differentiable vs non-differentiable methods.** Our work explicitly focuses on *differentiable* DAG learning methods. The non-differentiable pre- and post-processing indeed improves the results for all models (see description 'DAG structure' p.7 and the new Tab. 13, 14 in the appendix). However, non-differentiable processing and other methods (like CAM) have two important drawbacks: (1) they are not suited to gradient-based optimization commonly used in modern Deep Learning methods and (2) they cannot quickly sample DAGs (see explanation 'Processing time & Performance' in the rebuttal). Thus, the main advantages of DP-DAG is that it allows *fast and differentiable DAG sampling*.
>
> > It is unclear whether the DAG sampler is unbiased, i.e., it selects a DAG uniformly from the Markov equivalence class as number of samples goes to infinity.
>
> **Unbiased DAG sampler.** In theory, the optimum of ELBO loss in Eq.5 is the true posterior distribution over the DAGs i.e. $P(\mathbf{A} | \phi^*, \psi^*) \approx P(\mathbf{A} | \mathcal{D})$ where $\mathcal{D}$ denotes the dataset of observations. However, there is no guarantee and it is not desired that the posterior distribution $P(\mathbf{A} | \mathcal{D}) \propto P(\mathcal{D} | \mathbf{A}) P_{prior}(\mathbf{A})$ assigns the same probability to the Markov equivalence class. In particular, the graph $y \leftarrow x \rightarrow z$ might be assigned a  significantly higher probability than the graph $y \rightarrow x \rightarrow z$ if $x$ is a better predictor of $y$ than $y$ is a good predictor of $x$ given the observed data $\mathcal{D}$. This is a desired property since the parents relationships in VI-DP-DAG encode Granger-cause relationships (see Sec.4 'Variational inference with DP-DAG').

---

> > ### Comment · Reviewer_PQoz · 2021-11-28
> > **Thanks for addressing my comments**
> >
> > Thanks for addressing my comments. About the first comment, it is still possible to analyze the convergence rate of the proposed algorithm although it is hard to compare it with other related works if they did not provide convergence analysis. About the third comment, without further assumption on SCM, one cannot distinguish DAGs in an MEC even if we have access to the observational distribution. Thus, it is not a good property for a sampler to be biased in this scenario. Overall, I decided to keep my score unchanged.

---

> > > ### Author Response · Authors · 2021-11-29
> > > **Author Response to Reviewer PQoz (2)**
> > >
> > > Thanks a lot you for your comment.
> > >
> > > We propose a practical analysis of the convergence speed of VI-DP-DAG using direct time measurements (see Sec. 5.2 'Training time' p.9).. This comparison is particularly useful for real-world applications which have concrete time constraints. We are happy to make clear the difficulty of theoretical convergence analysis as suggested by the reviewer.
> > >
> > > We would like to clarify the bias property of VI-DP-DAG. If the observed data comes from e.g. the graph $x \rightarrow y$ with $y = 2 \pi \sin(x) + N(0, 0.01)$ and $x \in Unif([-2\pi, 2\pi])$, a neural network $f_\theta$ predicting $x \rightarrow y$ can achieve a significantly lower prediction error than a neural network predicting $y \rightarrow x$. In this case, this is desired to learn the direction $x \rightarrow y$ since $x$ is a better predictor of $y$ than $y$ is a good predictor of $x$. Hence, VI-DP-DAG explicitly assumes Granger-causality for edge direction (see Sec. 4 end of paragraph 'Variational inference with DP-DAG') and might not assigns in general the same probability to all DAGs in a MEC in *any* situations. Other works already mentioned that the causal direction can be inferred in some situations e.g. assuming non-Gaussian noise ([Dodge and Rosson, 2001](https://www.jstor.org/stable/2685529), [Shimizu and Kano, 2008](https://www.sciencedirect.com/science/article/abs/pii/S0378375808000918), [Shimizu et al., 2006](https://www.jmlr.org/papers/volume7/shimizu06a/shimizu06a.pdf)) or Gaussian SEM with equal error variances [Peters and Bühlmann, 2018](https://arxiv.org/pdf/1205.2536.pdf). We emphasize that we do not state that VI-DP-DAG can distinguish between DAGs within a MEC in *any* situations which is indeed known not to be possible in general ([Spirtes et al, 2000](https://link.springer.com/book/10.1007/978-1-4612-2748-9)). In particular, we explicitly mention that VI-DP-DAG performs better with undirected edges scores than with directed edges scores indicating that VI-DP-DAG might invert the edge directions (see Sec. 5.2, paragraph 'DAG structure').

---

### Decision · Program_Chairs · 2022-01-20

**Decision:**

Accept (Poster)

**Comment:**

The authors proposed an algorithm for sampling DAGs that is suited for continuous optimization. The sampling algorithm has two main steps: In the first step, a causal order over the variables is selected. In the second step, edges are sampled based on the selected order. Moreover, based on this algorithm, they proposed a method in order to learn the causal structure from the observational data. The causal structure learning algorithm is guaranteed to output a DAG at any time and it is not required any pre- or post-processing unlike previous work.

There were concerns by two reviewers on the slight lack of novelty ("the proposed method of this paper is only a combination of well-developed techniques") but I believe the proposed method is still worthwhile. In addition, the paper is overall well written and its experiment evaluation is thorough. It will be a nice addition to the field of differentiable causal discovery.

My recommendation is to accept the paper as a poster.